# Estimation of geostrophic current in the Red Sea based on Sea level anomalies derived from extended satellite altimetry data

Ahmed Mohammed Taqi[a,b], Abdullah Mohammed Al-Subhi[a], Mohammed Ali Alsaafani[a] and Cheriyeri Poyil Abdulla[a]

[a]Department of Marine Physics, King Abdulaziz University, Jeddah, Saudi Arabia;

[b]Department of Marine Physics, Hodeihah University, Hodeihah, Yemen

Correspondence to: Ahmed. M. Taqi (ataqi@stu.kau.edu.sa)

**Abstract**

Geostrophic currents data near the coast of the Red Sea has large gaps. Hence, the sea level anomaly (SLA) data of Jason-2 has been reprocessed and extended towards the coast of the Red Sea and merged with AVISO data at the offshore region. This processing has been applied to build a gridded data set to achieve best results for the SLA and geostrophic current. The results obtained from the new extended data at the coast are more consistent with the observed data (CTD) and hence geostrophic current calculation. The patterns of SLA distribution and geostrophic currents are divided into two seasons; winter (October – May) and summer (June – September). The geostrophic currents in summer are flowing southward all over the Red Sea except for narrow northward flow along the east coast. In winter, currents flow to the north for the entire Red Sea except for a small southward flow near the central eastern and western coast. This flow is modified by the presence of the cyclonic and anticyclonic eddies, which are more concentrated in the central and northern Red Sea. The results show anticyclonic eddies (AE) on the eastern side of the Red Sea and cyclonic eddies (CE) on the western side during winter. In summer, cyclonic eddies are more dominant for the entire Red Sea. The result shows a change in some eddies from anticyclonic during winter to cyclonic during summer in the north between 26.3°N –27.5°N. Furthermore, the lifespan of cyclonic eddies is longer than that of anticyclonic eddies.

# 1. Introduction

The Red Sea is a narrow semi-enclosed water body that lies between the continents of Asia and Africa. It is located between 12.5°N-30°N and 32°E-44°E in an NW-SE orientation. Its average width is 220 km and the average depth is 524 m (Patzert, 1974). It is connected at its northern end with the Mediterranean Sea through the Suez Canal and at its southern end with the Indian Ocean through the strait of Bab El- Mandab. The exchange of water through Bab El- Mandab (shallow sill of 137 m) is the most significant factor that determines the oceanographic properties of the Red Sea (Smeed, 2004).

During winter, the southern part of the Red Sea is subject to SE monsoon wind, which is relatively strong from October to December, with a speed of 6.7-9.3 ms$^{-1}$ (Patzert, 1974). During the summer season, the wind is shifting its direction to be from NW. On the other hand, in the northern part of the Red Sea, the dominant wind is NW all year around.

The circulation in the Red Sea is driven by strong thermohaline and wind forces (Neumann and McGill, 1961; Phillips, 1966; Quadfasel and Baudner, 1993; Siedler, 1969; Tragou and Garrett, 1997). Several studies in the Red Sea have focused on thermohaline circulation, where they found that the exchange flow between the Red Sea and Gulf of Aden consists of two layers in winter and three layers in summer through Bab El- Mandab (e.g.Phillips 1966; Tragou and Garrett 1997; Murray and Johns 1997; S. Sofianos and Johns 2015;Al Saafani and Shenoi, 2004; Smeed, 2004). Other studies describe the basin-scale circulation based on modelling approach, usually forced at a relatively low-resolution (1°x1°) by buoyancy flux and global wind (Clifford et al., 1997; Sofianos, 2003; Tragou and Garrett, 1997; Biton et al., 2008; Yao et al., 2014a,b). The horizontal circulation in the Red Sea consists of several eddies, some of them are semi-permanent eddies (Quadfasel and Baudner, 1993), that are often present during the winter (Clifford et al., 1997; Sofianos and Johns, 2007) in the northern Red Sea. The circulation system in the central Red Sea is dominated by cyclonic (CE) and anticyclonic eddies (AE), mostly between 18°N and 24°N. Eddies are also found in the southern Red Sea but not in a continuous pattern (Johns et al., 1999). Zhan et al., (2014) reported recurring or persistent eddies in the north and the central Red Sea, although there are differences in the number of eddies, their location, and type of vorticity (cyclonic or anticyclonic).

The long-term sea level variability in the Red Sea is largely affected by the wind stress and
the combined impact of evaporation and water exchange across the strait of Bab El Mandeb
(Edwards, 1987; Sultan et al., 1996). The Sea level in the Red Sea is higher during winter and
lower during summer (Edwards, 1987; Sofianos and Johns, 2001; Manasrah et al., 2004). It is
characterized by two cycles, annual and semi-annual, where the annual cycle is dominant
(Abdallah and Eid, 1989; Sultan and Elghribi, 2003).
In recent years, there has been an increasing interest for using satellite altimetry Sea level
anomaly (SLA) which offer large coverage and long data period for providing measurements of
SSH, wave height and wind speed (Chelton et al., 2001). However, the altimeter data undergoes
several processing stages for corrections due to the atmosphere and ocean effects (Chelton et al.,
2001).  The satellite altimetric data has been used for the open ocean for a long time with great
success, while the data of the coastal region suffers from gaps of almost 50 km from the coastline.
The coastal region requires further corrections due to additional difficulties based on the closeness
of the land (Deng et al., 2001; Vignudelli et al., 2005; Desportes et al., 2007; Durand et al., 2009;
Birol et al., 2010). In the past two decades, many researchers have sought to develop different
methods to improve the quality, accuracy and availability of altimetric data near the coast (e.g.
Vignudelli et al., 2000; Deng and Featherstone, 2006; Hwang et al., 2006; Guo et al. 2009, 2010;
Vignudelli et al., 2005; Desportes et al., 2007; Durand et al., 2009; Birol et al., 2010; Khaki et al.,
2014; Ghosh et al., 2015; Taqi et al., 2017).  The satellite altimetry faces three types of problems
near the coast; (1) the echo interference with the surrounding ground as well as the inland water
surface reflection (Andersen and Knudsen, 2000; Mantripp, 1966), (2) environmental and
geophysical corrections such as dry tropospheric correction, wave height, high frequency and tidal
corrections from global models, etc. and (3) spatial and temporal corrections during sampling
(Birol et al., 2010).

The ocean currents advect water worldwide. They have significant influence on the transfer
of energy and moisture between the ocean and the atmosphere. Ocean currents play a significant
role in climate change in general. In addition, they contribute to the distribution of hydrological
characteristics, nutrients, contaminants and other dissolved materials between the coastal and the
open areas, and among the adjacent coastal regions. Ocean currents carry sediment from and to
the coasts, so play a significant role in shaping of the coasts. That is important densely inhabited

coastal region, producing large amounts of pollutants. Understanding of the currents helps us in
dealing with the pollutants and coastal management.

The objective of the present research is to study the geostrophic current in the Red Sea
including the coastal region using the modified along track Jason-2 SLA along the coast produced
by Taqi et al., (2017).

## 2.   Material and Methods

### 2.1. Description of data

#### 2.1.1 Fourier series model (FSM) SLA

The SLA data used in this study is weekly Jason-2 along the track from June 2009 (cycle
33) to October 2014 (cycle 232) which has been extended to the coastal region by Taqi et al.,
(2017). To cover all the period, additional tracks were added up to December 2014 (cycle 239).
The extended data shows a good agreement with the coastal tide gauge station data. In brief, the
FSM method of extending SLA consists of four steps; the first step is the removal from SLA the
outliers which are outside three times standard deviation from mean. Second step; the SLA is
recomputed using Fourier series equation along the track.  Third step; the data is then filtered to
remove the outliers in the SLA with time similar to the first step. Finally, the SLA data is linearly
interpolated over the time to form the new extended data which is called FSM. For more details
on the FSM method, refer to Taqi et al., (2017).

#### 2.1.2 AVISO, Tide Gauge, and hydrographic datasets

This study uses two types of SLA data; The first set is the  (SLA), which has been
downloaded from the Archiving Validation and Interpretation of Satellite Oceanographic (AVISO)
(ftp://ftp.aviso.altimetry.fr/global/delayed-time/grids/msla/all-sat-merged). The second dataset is
the SLA from the extended FSM data. The temperature and salinity profiles used for geostrophic
estimation are received from three cruises, the first cruise was during March 16 to 29, 2010
onboard R/V Aegaeo between 22°N to 28°N along the eastern Red Sea with a total of 111
Conductivity, Temperature and Depth (CTD) profiles. For more details; see Bower and Farrar
(2015). The second cruise was on April 3 to 7, 2011 onboard Poseidon between 17°N to 22°N in
the central eastern Red Sea and the third one was during October 16 to 19, 2011 onboard the same
vessel between 19°N to 23°N in the central eastern Red Sea as a part of Jeddah transect, KAU-
KEIL Project.  For more details; consult R/V POSEIDON cruise P408/1 report (Schmidt et al.,

2011). The availability of in-situ observations is limited in space and time because of the spatial
and temporal distribution of the available cruises. Finally, three tide gauges data at the eastern
coastline of the Red Sea are obtained from the General Commission of Survey (SGS) at the
Kingdom of Saudi Arabia (Fig.1) and their location details are shown in Table 1.
**2.2 Method**

The SLA data used in this study are coming from two sources: (1) the FSM data near the

coast and (2) the AVISO data along the axis of the Red Sea. The steps to merge the two datasets
and calculating the geostrophic currents are given below.

First, the along-track FSM data are used to produce gridded data to a spatial resolution of

$0.25° \times 0.25°$ for the comparison with Aviso data. In the second step, AVISO data near the coast
is removed, and replaced with the coastal FSM gridded data leaving space between the two data
set according to the width of the sea: either one or two grid cells. This gap was filled using kriging
interpolation method to smooth the dataset. The merged data hereafter called as FSM-SLA.
Finally, surface geostrophic currents are estimated from FSM-SLA data using the following
equation;

$$u_g = -\frac{g}{f}\frac{\partial \zeta}{\partial y} \qquad\qquad v_g = \frac{g}{f}\frac{\partial \zeta}{\partial x} \qquad (1)$$

Where $(u_g, v_g)$ is the surface geostrophic current, $g$ is gravity, f is the Coriolis parameter and $\zeta$ is
the sea surface height. The estimation of geostrophic currents from CTD data is using the following
equation;

$$u_g = -\frac{1}{f\rho}\frac{\partial p}{\partial y} \qquad\qquad v_g = \frac{1}{f\rho}\frac{\partial p}{\partial x} \qquad (2)$$

where ρ is the density of seawater, $p$ is hydrostatic pressure derived from the density. The stations
have depths that vary from 50 to 2344 m. However, most of the stations (~90 %) exceed the 500
m depth. Previous study by Quadfasel and Baudner (1993) used 400 m as level of no motion to
calculate geostrophic current in the Red Sea. Based on ADCP measurements, Bower and Farrar
(2015) shown that, on average, 75–95 % of the vertical shear occurred over the top 200 m of the
water column. Moreover, the ADCP measurements of current speed below 500 m is very small;
about ~0.06 m/s at 600 m depth (Bower and Farrar, 2015). Therefore, expecting negligible
variability below 500 m, a depth of 500 m was selected as a level of no motion. We have compared
the geostrophic current corresponding to level of no motion at 500m and 700m. The observed
difference between both are negligibly small, with RMSE around 0.003 m/s at surface.
Table 1. The location of tide gauge stations and period of measurement.

| Station | Latitude | Longitude | Period |
|---------|----------|-----------|--------|
| Jazan | 16.87 | 42.55 | 1/1/2013 to 31/12/2013 |
| Jeddah | 21.42 | 39.15 | 1/1/2013 to 31/12/2013 |
| Yanbu | 23.95 | 38.25 | 1/1/2013 to 31/12/2013 |


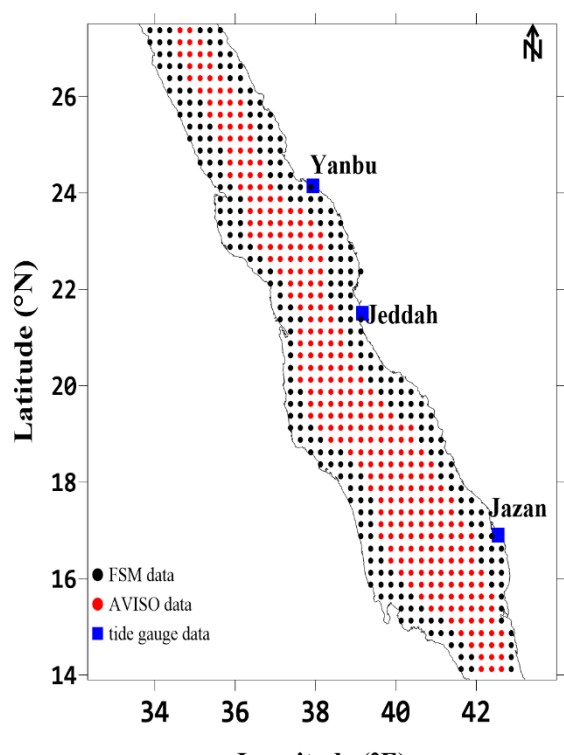

Figure 1. show the study area and the grid-points locations with a spatial resolution of 0.25° ×
0.25° and locations of the tide gauges.
**3.  Result and Discussion**
**3.1 Validation of FSM-SLA and geostrophic current**

The statistical analysis has been conducted to show the quality of FSM-SLA as compared

with AVISO. The Correlation Coefficient (CC) reveals a good agreement between the two
datasets in the open sea (about 0.7 to 0.9) and is shown in Fig. 2. In contrast, near the coasts,
weak correlation is found between the two datasets, the correlation coefficient being 0.45 to 0.7.
Furthermore, the observed SLA from the coastal tide gauge is compared with the FSM-SLA data
and AVISO datasets. Table 2. illustrates some of the statistical analysis, where the root mean
square error (RMSE) is less for FSM-SLA as compared to that of AVISO.

Table 2. statistical analysis for AVISO and FSM-SLA data with observed coastal tide gauge data
( in 2013).

|  | Jasan | | Jeddah | | Yanbu | |
|---|---|---|---|---|---|---|
|  | FSM-SLA | AVISO | FSM-SLA | AVISO | FSM-SLA | AVISO |
| CC | 0.936 | 0.914 | 0.915 | 0.906 | 0.907 | 0.895 |
| RMSE(m) | 0.073 | 0.085 | 0.069 | 0.094 | 0.067 | 0.104 |
| Note: The p-value corresponding to all comparison is very low (P<0.0001), indicating that the results from correlation are significant. | | | | | | |


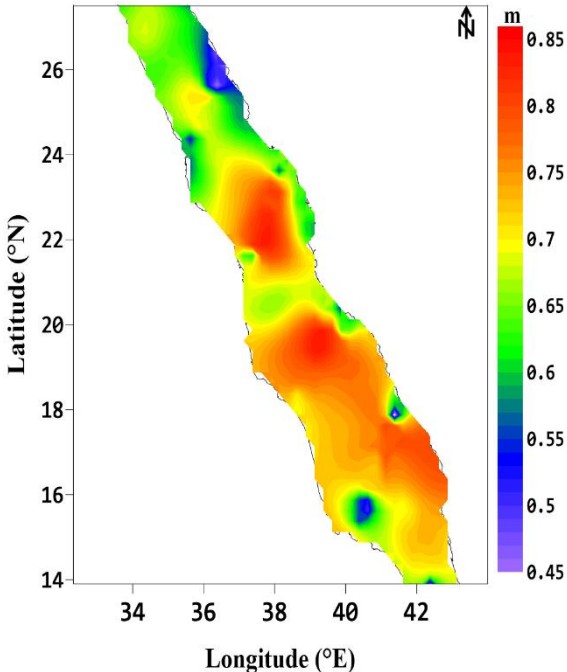


Figure 2. show the correlation coefficient between AVISO and FSM data

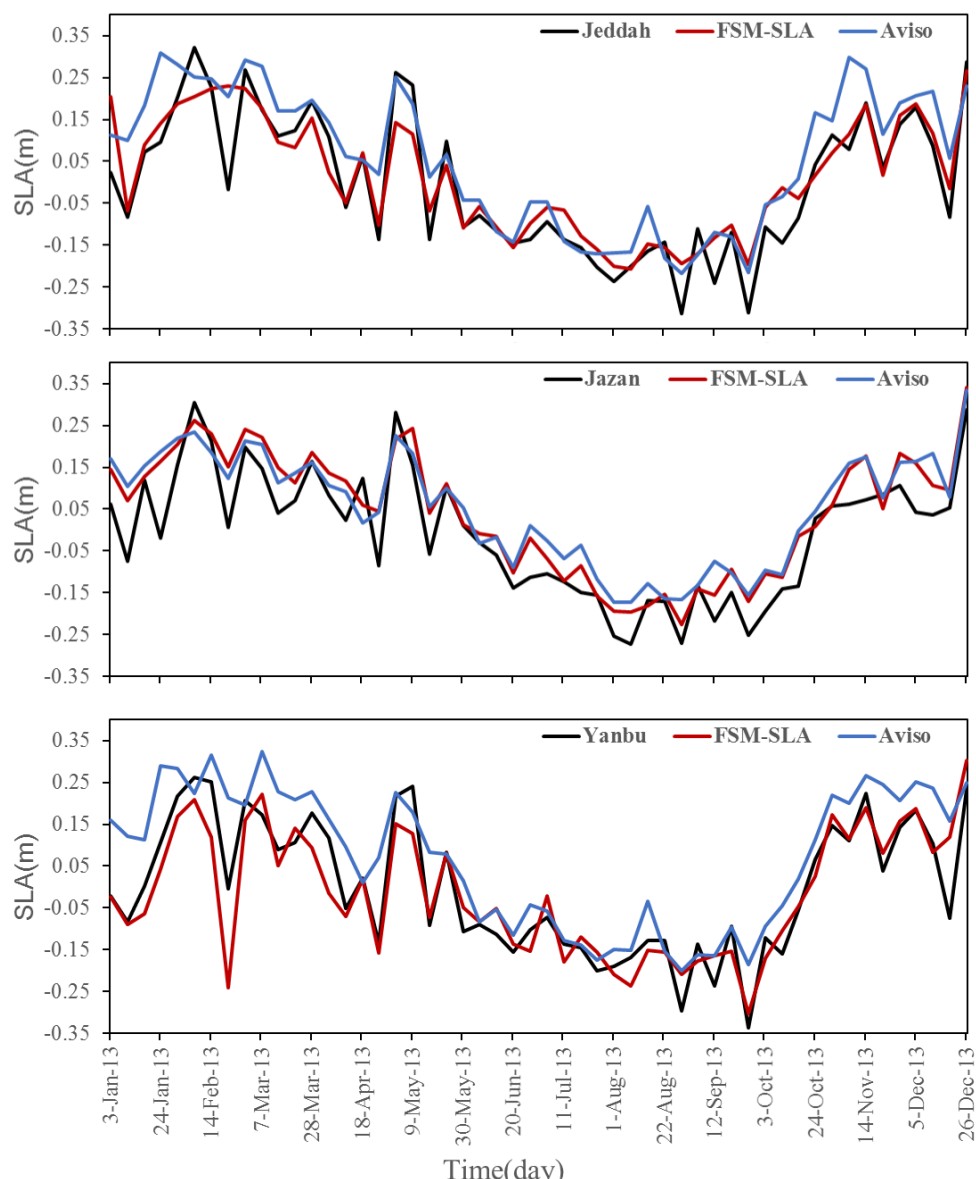

Figure 3. Comparison of SLA from three tide gauge (black), with grid FSM-SLA data (red) and Aviso (blue)

Figure 3 shows the SLA time series for 2013 from the three coastal stations as compared with the FSM-SLA and AVISO. The three stations datasets have similar seasonal pattern and FSM-SLA coincides with observed SLA in shorter-duration fluctuations. The comparison of FSM-SLA data and the observed SLA data (at Jazan, Jeddah, and Yanbu stations) shows a better correlation than between the AVISO and observed SLA data as shown in Fig.3 and Table .2. These correlation coefficient differences indicate that the FSM-SLA shows better accuracy near the coast. These

results were consistent with those obtained for along-track Jason-2 SLA with coastal stations by
Taqi et al., (2017).

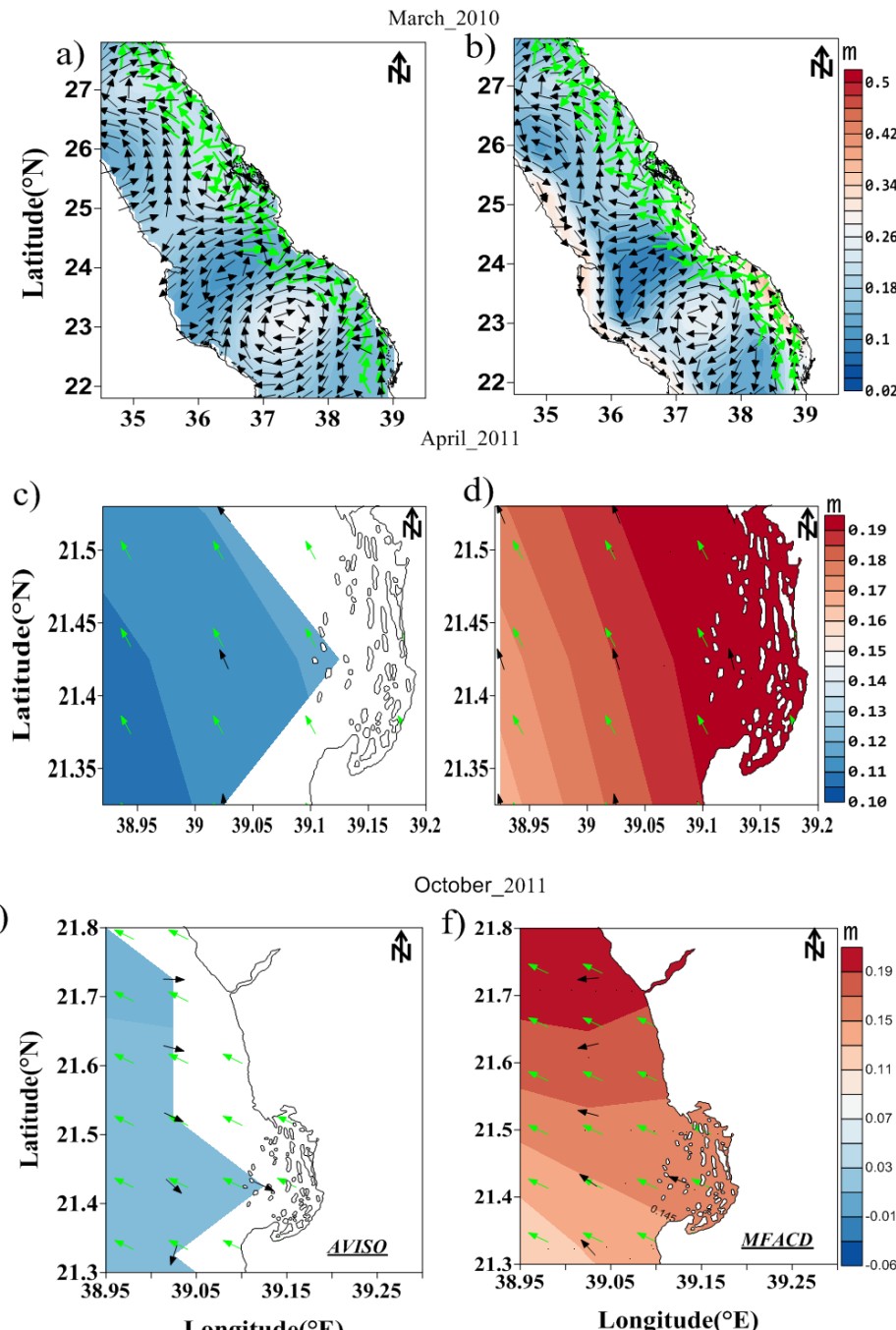

Figure 4. Comparison for three months' SLA (color) and geostrophic currents (black vectors)
between (left) AVISO and (right) FSM-SLA and Green vectors show geostrophic currents from
CTD data.

Figure 4 shows a comparison between the geostrophic currents for the central Red Sea

derived from AVISO and FSM-SLA for three different times (March 2010, April 2011, and
October 2011), those different periods corresponding to the timing of three cruises described in
section 2.1.

It can be seen from the Fig. 4(b, d & f) that there is a significant matching in the directions

of geostrophic currents from FSM-SLA with CTD data near the coast and offshore. This result is
in agreement with Bower and Farrar (2015) findings, especially in October 2011 ( Fig. 4f). In
March 2010, the geostrophic current near the coast estimated from FSM-SLA is in match with
directions of CTD-derived geostrophic current in most regions. However, the directions of
geostrophic currents from AVISO are not always in match with CTD-derived one especially in
October 2011.

In March 2010 the geostrophic currents along the eastern coast of the Red Sea are towards

the north for both FSM-SLA and AVISO, except between 22.2˚ – 23˚N, where the FSM-SLA and
CTD data geostrophic currents are in the same direction while AVISO geostrophic current is in
the opposite direction (see Fig. 4a,4b).
Table 3. statistical analysis for the speed of geostrophic current from FSM-SLA and AVISO
compared with CTD-derived geostrophic current from the three cruises.

| | | Month-Year | Bias (m/s) | RMSE (m/s) | CC | P-Value |
|---|---|---|---|---|---|---|
| current speed | FSM-SLA | Mar-2010 | -0.0085 | 0.065 | 0.54 | 0.01 |
| | AVISO | | -0.01 | 0.08 | 0.48 | 0.14 |
| | FSM-SLA | Apr-2011 | -0.28 | 0.31 | 0.61 | 0.02 |
| | AVISO | | -0.87 | 0.89 | 0.44 | 0.13 |
| | FSM-SLA | Oct-2011 | -0.19 | 0.49 | 0.53 | 0.10 |
| | AVISO | | -0.51 | 0.70 | 0.49 | 0.16 |


The speed of geostrophic current data derived from FSM-SLA and CTD during the months

(March 2010, April 2011, October 2011) shows a stronger correlation compared with the speed of
geostrophic current derived from AVISO and CTD as shown in Fig. 4 and Table 3.

### 3.2 Description of FSM-SLA and geostrophic current

Figure 5 shows monthly climatology variation for the 6-year period for SLA and geostrophic current. The SLA is higher during the period from October to May and lower during rest of the year, this pattern is consistent with previous studies ( Patzert, 1974; Edwards, 1987; Ahmad and Sultan, 1989; Sofianos and Johns, 2001; Sultan and Elghribi, 2003;Manasrah et al., 2004, 2009). Based on calculations made here, the geostrophic current of Red Sea along the eastern coast is northward while along the western coast is southward. This northward flowing current is consistent with a previous study by Bower and Farrar (2015). Similar results are also obtained from three-dimensional modeling by (Clifford et al., 1997; Eshel and Naik, 1997; Sofianos, 2003, 2002). The Fig.5 presents the surface circulation during January in the northern part, where two eddies formed between $25^\text{o}$ – $27.5^\text{o}$N. The first eddy is an anticyclone between $26.3^\text{o}$ – $27.5^\text{o}$N on the eastern side of the Red Sea. The other eddy is cyclonic located between $25^\text{o}$ – $26.3^\text{o}$N near the western coast. To the south of that, there are two other eddies between $22.5^\text{o}$ – $24.7^\text{o}$N, cyclonic on the western side and anticyclonic on the eastern side. These results match those observed in previous studies by (Eladawy et al., 2017; Sofianos and Johns, 2003a). Two cyclonic eddies and an anticyclonic eddy found at $19.5^\text{o}$ – $22.5^\text{o}$N are consistent with those modeled by Sofianos and Johns, (2003). Near Bab al-Mandab, there is a cyclonic eddy on the western side between $15^\text{o}$ – $16.5^\text{o}$N.

In February, the surface circulation of the Red Sea is similar to that during January, with some differences in the eddies structure. The anticyclonic eddy near $27^\text{o}$N on the eastern sides of the Red Sea starts shifting toward the western coast, while a cyclonic eddy at $25^\text{o}$ – $26.3^\text{o}$ N starts appearing. The cyclonic eddies between $22.5^\text{o}$ – $24.7^\text{o}$N on the western side are less clear in this month.

In March and April, all the eddies are located along the central axis of the Red Sea. In the north, the anticyclonic eddy near $27^\text{o}$N is shown in both months, while the cyclonic eddy is not clear during March and April. The anticyclonic eddy shown near 23-24$^\text{o}$N during March is weakening during April.  Also, the anticyclonic eddy between 19-20$^\text{o}$N is shrinking during April.

In May, there is no clear eddy between $27.5^\text{o}$N and $25^\text{o}$N. However, four eddies are clearly existing between $19.5^\text{o}$ –$25^\text{o}$N; two cyclonic eddies at $24^\text{o}$ – $25^\text{o}$N, and $20^\text{o}$ – $22^\text{o}$N, two anticyclonic eddies at $23^\text{o}$ – $24^\text{o}$N, and $19.5^\text{o}$ – $20^\text{o}$N. From the previous results, it can be seen several cyclonic

and anticyclonic eddies distributed all over the Red Sea and these results match those in modelling
studies (Clifford et al., 1997; Eladawy et al., 2017; Sofianos, 2003, 2002, Yao et al., 2014a)
During June, the flow of the geostrophic currents in the northern part reversed its direction.
This accompanies a formation of large cyclonic eddy extending from $25.5^o$– $27.5^o$N occupying the
entire width of the Red Sea. To the south of it, another cyclonic eddy observed between $24^o$ – $25^o$N
and an anticyclonic eddy between $23^o$ – $24^o$N are also noticed during June with a similar strength
during May.  The cyclonic eddy seen between $17^o$ – $20^o$N during May, is also seen during this
month with more strength. To the south of it, the flow is towards the Bab el-Mandab following
normal summer pattern. The flow pattern along the coast is similar to results of (Chen et al., 2014)
for winter (January to April). The short-term climatology of geostrophic current in the Red Sea is
dominated by cyclonic and anticyclonic eddies all over the Red Sea, and especially in the central
and northern parts of the sea.
During July-September, the flow of the geostrophic currents structure is similar to that of
June with two cyclonic eddies north of $24.5^o$N and an anticyclonic eddy between $23^o$ – $24^o$N.  South
of these eddies, another cyclonic eddy extends to $19^o$N.  Furthermore, south of $19^o$N, there is an
outflow towards the south over almost all the width of the Red Sea with narrow inflow along the
eastern coast of the Red Sea. The Fig. 6 also shows an anticyclone between $18°$-$19^o$N and a cyclone
between $16°$-$17^o$N during August and September. These results are consistent with the results from
previous studies (Clifford et al., 1997; Eladawy et al., 2017; Sofianos, 2003, 2002, Yao et al.,
2014b).
During summer (June-September), the changes in wind speed and direction cause reversals
of the direction of flow consequently, the locations of eddies are also changed (Chen et al., 2014).
The surface current flows from the Red Sea to the Gulf of Aden through the Bab-el-Mandeb. The
anticyclonic eddy shown in the north at $27.5^o$N in winter is replaced with cyclonic eddy, during
this season. In summer is dominated by cyclonic eddies as shown in Fig. 6.

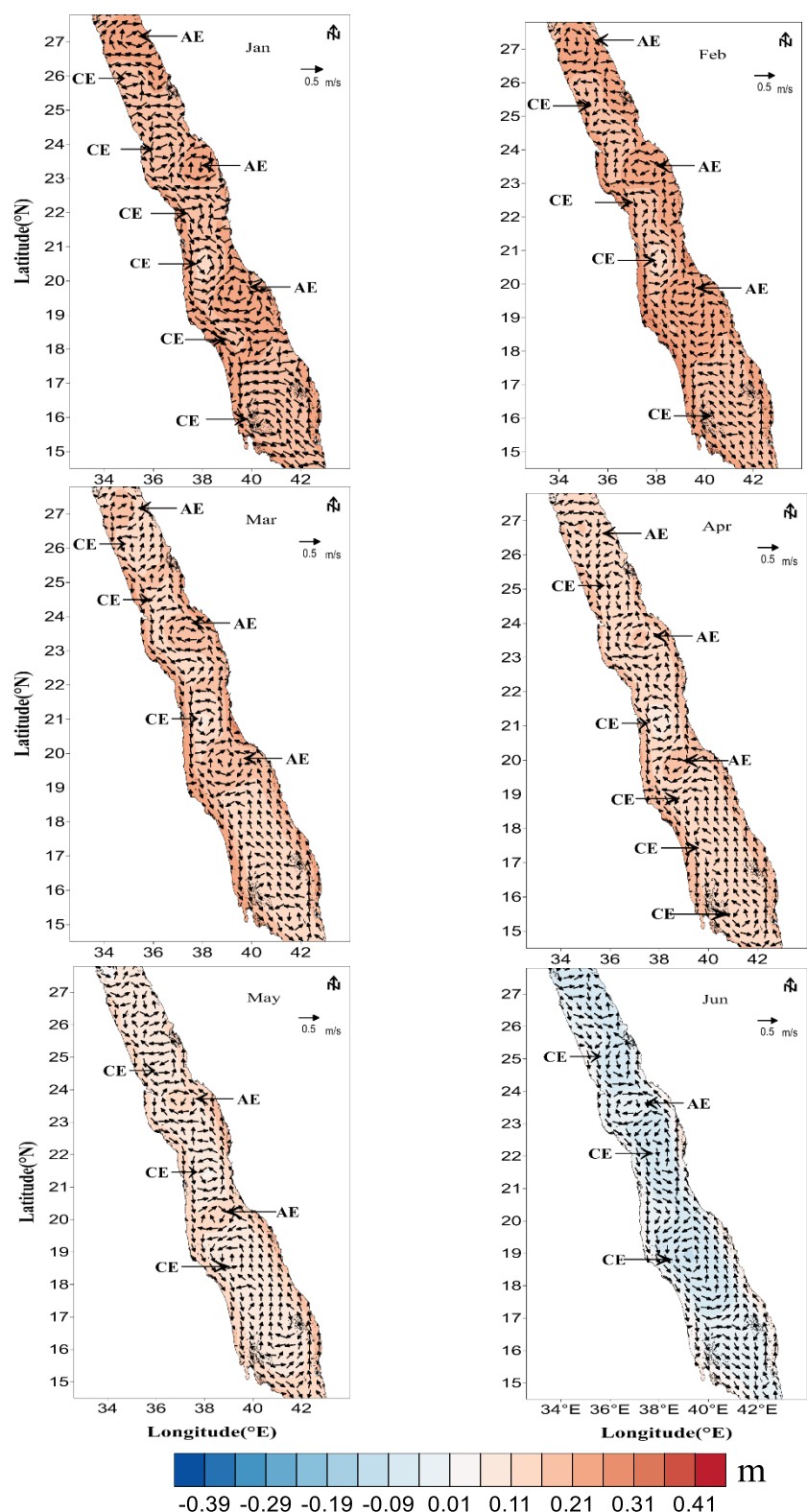

Figure 5. shown monthly climatology for geostrophic current and Sea level anomaly (Reference current length =0.5m/s)

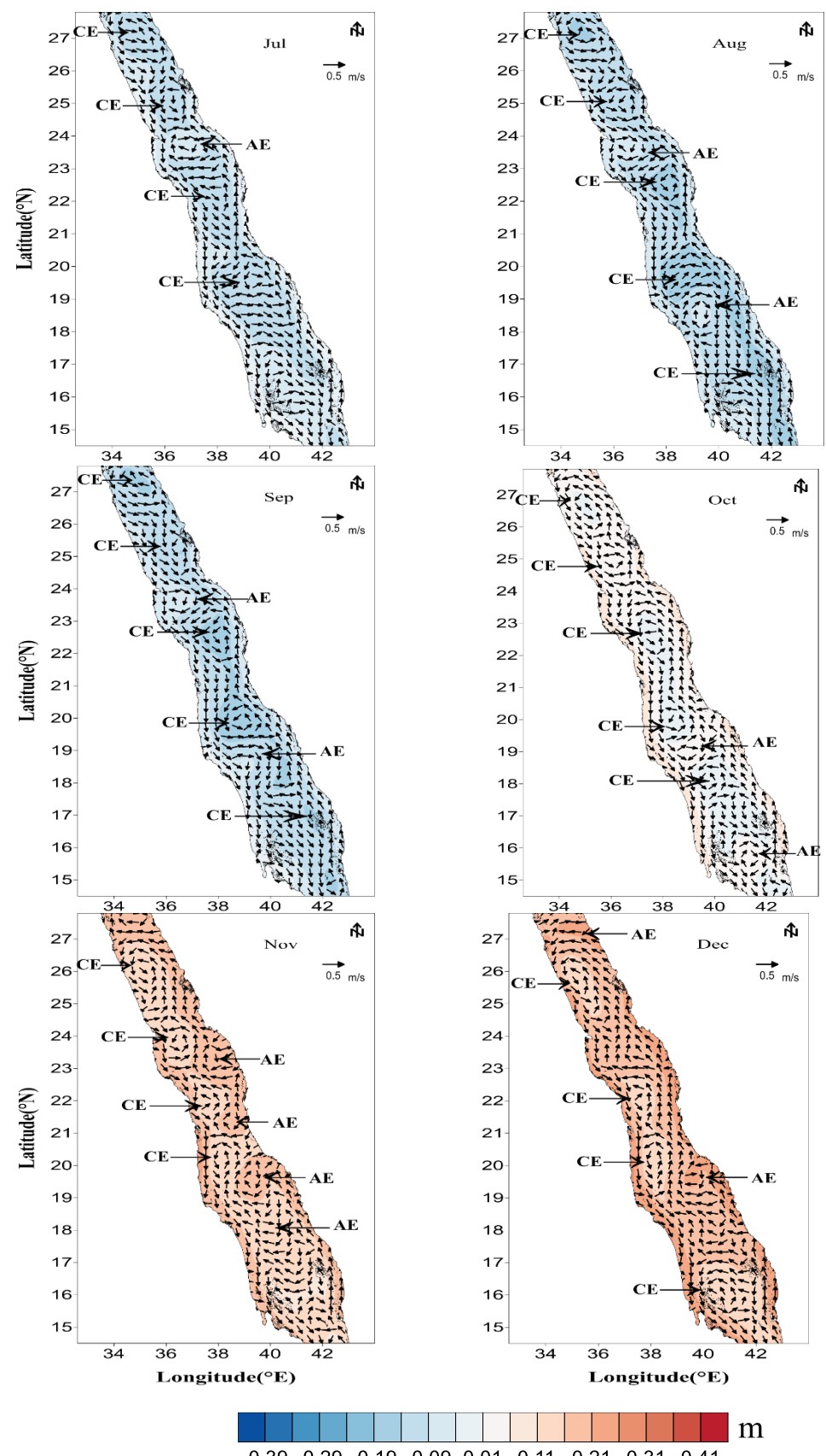

Figure 6. As Fig. 5 for July to December
During October, the geostrophic current is weak as compared with that during September,
still cyclonic but with less strength. The anticyclone seen during September between 23º – 24ºN is
not clear during October but an anticyclonic eddy forms between 15ºN-16ºN.  In the central and
southern parts, the flow of the geostrophic currents is towards south along the western coast and
towards the north along the eastern side with the presence of cyclonic and anticyclonic eddies in
the central axis of the Red Sea with a weak flow. In November and December, the structure of
geostrophic currents are similar to that of October but with stronger currents and well established
cyclonic and anticyclonic eddies.
During early summer the eddies are concentrated along the central Red Sea. By August
and September some of the cyclonic eddies are shifted towards the eastern coast. During winter
the cyclonic eddies are often condensed along the western side of the Red Sea, while anticyclonic
eddies along the eastern side of the Red Sea. Their formation might be related to wind forces and
thermohaline (Neumann and McGill, 1961; Phillips, 1966; Quadfasel and Baudner, 1993; Siedler,
1969; Tragou and Garrett, 1997).
Since the general circulation in the Red Sea is largely modified with the presence of
cyclonic and anticyclonic eddies, the identification of eddies in the study area were conducted
based on defining the eddies in terms of SLA (Chelton et al., 2011). Figure 7 shows statistical
variability of lifespan, number of eddies, amplitude, and the mean speed of geostrophic current in
the center of the eddies with latitude for 6 years. Statistical analysis indicates that eddies are
generated over the entire Red Sea, mostly concentrated between 18°-24ºN, obviously stronger than
any other latitude. The amplitude of an eddy has been defined as the differences between the
estimated basic height of the eddy boundary and the extremum value of SLA inside the eddy
interior parts. The mean amplitude for anticyclonic is between 1.3 cm at southern Red Sea to 5.3
cm at northern Red Sea and for cyclonic eddy is between 1.6 cm at southern Red Sea to 4.2 cm at
northern Red Sea. The result indicates the average value of eddy amplitude in the Red Sea
(including low latitude and high latitude) is about 2.96 cm, which is within the reasonable range
defined by (Chelton et al., 2011). The average lifespan of the cyclonic eddies is longer than that of
the anticyclonic eddies. Moreover, the mean speed of geostrophic current for the entire Red Sea is
about 5-10 cm/s, which has reached three-times higher in the 25°-26ºN latitude band for both

cyclonic and anticyclonic. These results match those observed in previous study Zhan et al.,
(2014).

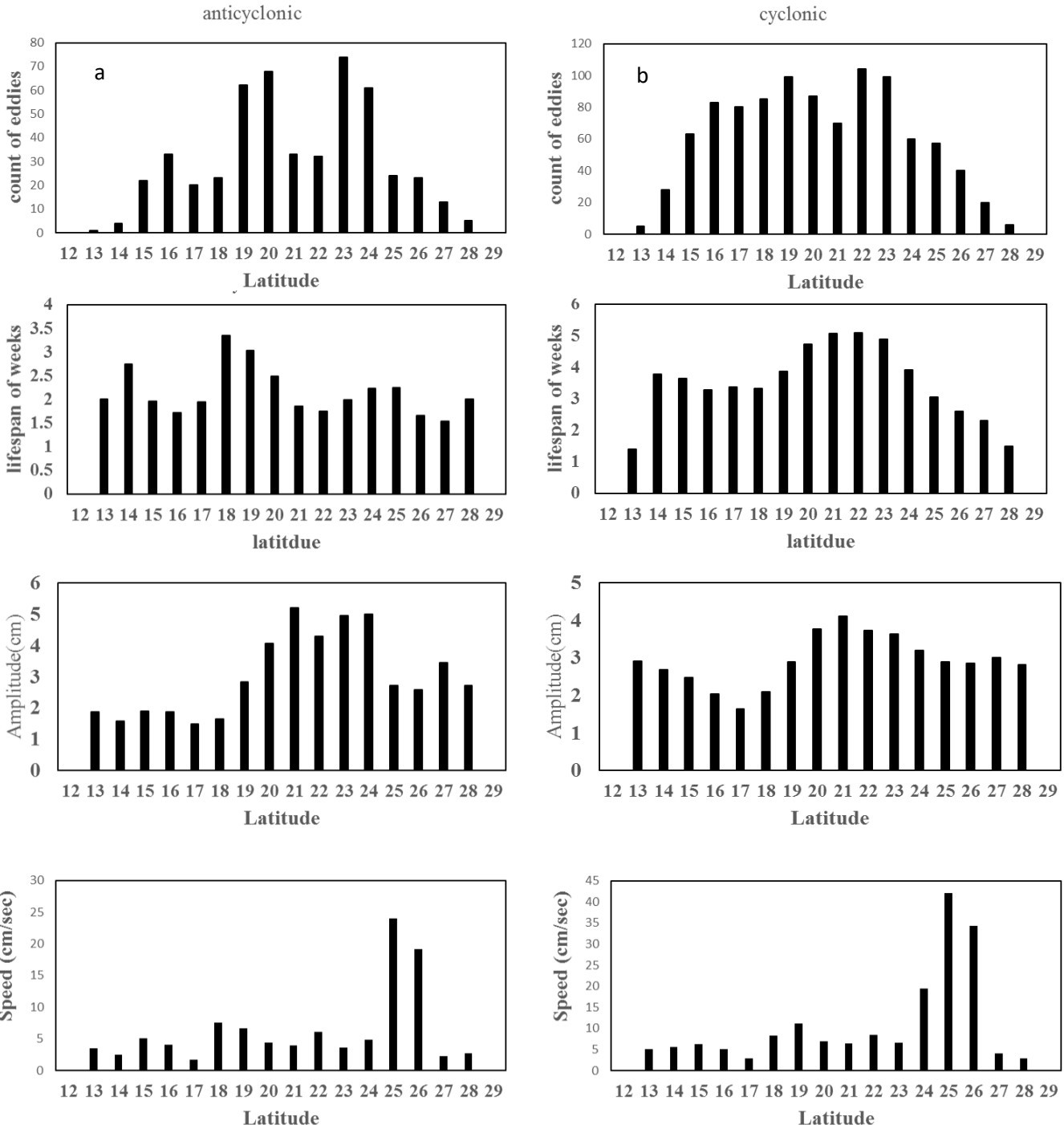

Figure 7. the variability of eddies with latitude for cyclonic (right panel) and anticyclonic (left
panel).

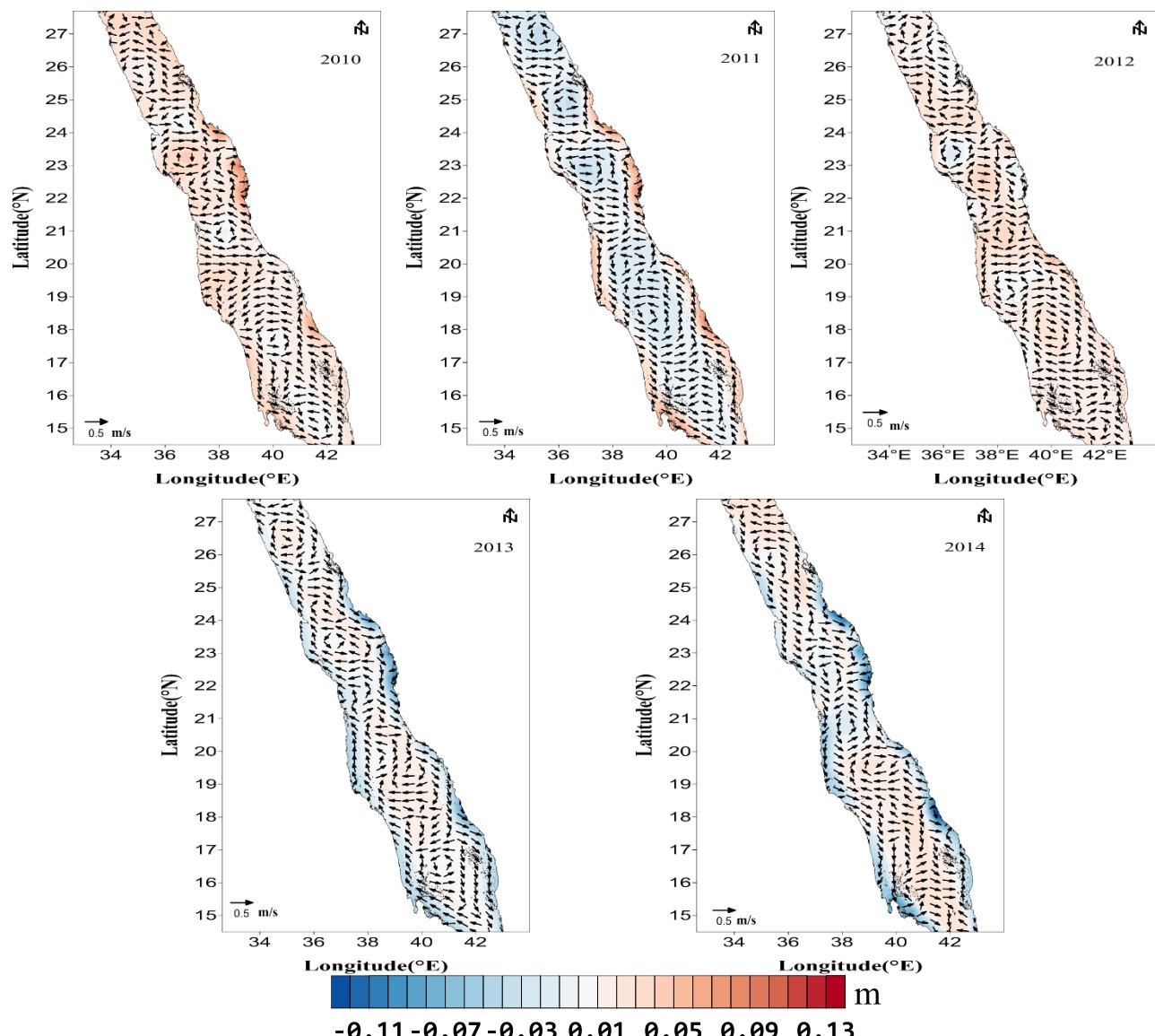

Figure 8. Maps of the annual mean SLA as a deviation from 6-yr mean.

Figure 8 shows the annual mean of SLA as deviation from the 6-year mean. The interannual variability of SLA and geostrophic currents is clearly seen in the southern part of the Red Sea while in the northern part, the pattern is similar for all years except for 2013 where the cyclonic eddy is replaced by anticyclonic eddy. The SLA and geostrophic distribution observed during 2011 are similar to that shown in Papadopoulos et al., (2015), with the cyclonic eddy along the eastern side seen more clearly. Moreover, due to extension of our data we could compute the cyclonic pattern up to the coast. The geostrophic currents direction is irregular along the coast but is northward most of the time. The eddies were mostly concentrated in the north and central parts of the Red Sea.

The statistical analysis between annual FSM-SLA with 6-year climatology shown in

Table 4. The correlation is significant for all the years with standard deviation(σ) less than 0.1.
the Bias is very small regardless its sign.
Table 4. Statistical analysis for the annual mean of FSM-SLA compared deviation from 6-yr
mean.

| Year | Bias | σ | CC |
|------|------|------|------|
| 2010 | -0.012 | 0.034 | 0.544 |
| 2011 | -0.009 | 0.023 | 0.774 |
| 2012 | -0.010 | 0.025 | 0.548 |
| 2013 | -0.011 | 0.033 | 0.791 |
| 2014 | -0.019 | 0.047 | 0.726 |

Note: The p-value corresponding to all comparison is very low (P<0.0001), indicating that the results from correlation are significant.

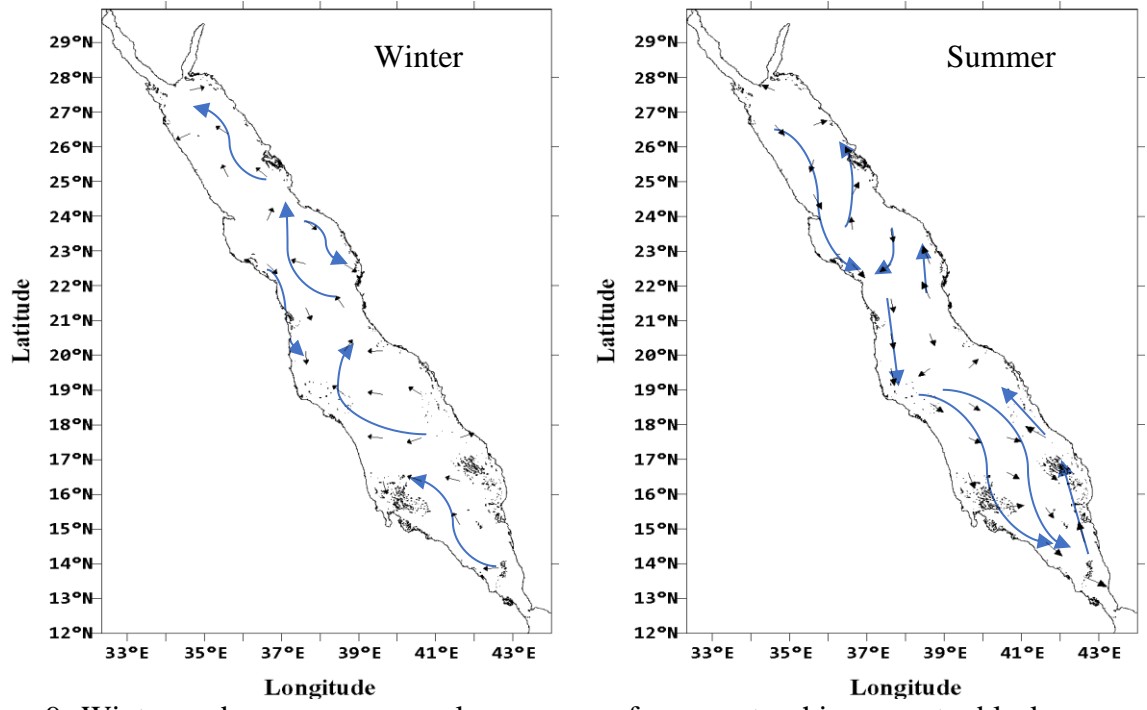

Figure 9. Winter and summer seasonal average surface geostrophic currents, black arrows are
actual surface geostrophic currents and blue arrows are schematic streamline.
Figure 9 shows the general schematic of the seasonal variability of geostrophic currents derived
from 6 years. During winter, the mean flow is toward the north all over the entire width of Red
Sea, this result agrees with Sofianos and Johns, (2003).  In addition, a southward flow of
geostrophic currents were observed along the eastern coast at (22°N-24°N) and the western coast
at (23°N-20°N). During summer, the flow is towards the south along the western side of the sea
while in southern part the flow spreads across most of the width of the Red Sea with a narrow
northward flow near the eastern coast.
**4. Conclusion**
In general, the geostrophic current has been estimated from FSM-SLA for Red Sea region,
and the distribution of the geostrophic current shows that the winter period extends from October
to May and summer period extends from June to September. This pattern is similar to that shown
by (Sofianos and Johns, 2001). There was a lack in measurements of coastal currents in the Red
Sea. This study was able to produce data near the coast. The major new findings from the present
study include the monthly geostrophic pattern in the Red Sea which has not been published before.
The southern Red Sea shows significant interannual variability in the geostrophic current
pattern, while the central and northern parts show negligible difference over the years. The
geostrophic current flow along the eastern coast is towards the north while along the western coast
of the sea it is southward. Seasonally, the geostrophic currents in summer are flowing southward
except along the eastern coast where they flow in the opposite direction. In winter, currents flow
to the north for the entire sea except for a southward flow along small part of the eastern (22°N-
24°N) and western coast (20°N-23°N). In this study, northward flowing eastern coastal current
during summer is documented for the first time in the Red Sea.
The cyclonic eddies were relatively larger than the anticyclonic eddies in the Red Sea. The
eddies are concentrated in the central and northern Red Sea more than in the southern side. The
mean amplitude for anticyclonic and cyclonic eddies at lower latitudes have small amplitude and
at higher latitudes have a larger amplitude. In winter, the cyclonic eddies are beside the western
coast and anticyclonic eddies on the eastern side in the Red Sea, while in summer it is concentrated
along the central Red Sea in early summer, with some cyclonic eddies transfer to the east coast in
late summer. Also, in some locations there is a noticeable change from anticyclonic during winter
to cyclonic during summer and vice versa between 26.3°N –27.5°N. The analysis of the eddies
found that during the summer the cyclonic eddies are dominant in the entire Red Sea, while eddies
of both polarities observed during winter.  The finding of this paper is considered the first of its
type in the Red Sea for extending SLA and geostrophic currents to the coast beside giving more
details of eddies spatial and temporal variabilities in the coastal region.

## Acknowledgments

The authors are deeply grateful to the data providers. JPL Physical Oceanography Distribution Active Archive Center (PODAAC), the Archiving Validation and Interpretation of Satellite Oceanographic (AVISO) to provide Jason-2 data. This also extends to the Saudi Arabian GCS for providing hourly tide gauge data along the coast of the Red Sea. They are thankful for the High-Performance Computing center at King Abdulaziz University (http://hpc.kau.edu.sa) for giving us a chance to use their facilities during analyses of data. Our thanks are for King Abdulaziz University, Jeddah, Saudi Arabia and Hodeidah University, Yemen for making this research possible.

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
