# Peer review of "Estimation of geostrophic current in the Red Sea based on Sea level anomalies derived from extended satellite altimetry data"

_Ocean Science, 2018_

## Referee Comment (RC1) · Anonymous Referee #1 · 26 Sep 2018

The paper "Estimation of geostrophic current in the Red Sea based on Sea level anomalies derived from extended satellite altimetry data" by Taqi et al. focuses on describing the geostrophic currents and eddy field in the Red Sea based on altimetry data, extended to the coast using a method proposed by same authors (Taqi et al., 2017). The first part consists of a continuation of the validation of the method (adding hydrographic data for estimating the geostrophic velocity) and the second part provides an analysis of the monthly climatology of the sea level anomaly (SLA) and the corresponding surface currents (averaging 6 years satellite data).

The validation part provides very little additional analysis compared to the Taqi et al.,

2017, while there is no information and/or reference related to the cruises that were used for estimating the geostrophic currents (lines 107-110). Actually, after checking the reference provided later in the text (e.g. Bower and Farrar, 2015) the cruise(s) covered a much larger area than the one used and shown in this paper. It is not understood why the authors selected the specific regions to perform the validation. The cruises also used an LADCP and thus the adoption of 700 m reference level seems arbitrary (actually most of the stations are shallower than that). The comparison and error estimation is very qualitative (comparing figures) and in figure 4a,b (the largest area covered) it is impossible to visualize the results.

The second part is very weak, merely describing the twelve monthly SLA/geostrophic velocity figures. The methodology of averaging 6 years of SLA data to describe the climatology of the complex Red Sea eddy field is not appropriate. While the basin-scale seasonal variability of the SLA can benefit very little from the new method of extending the data to the coast (this comparison is not shown), the averaging could mask the eddy field and produce artificial features. More advanced methods, including the interannual variability of the SLA/geostrophic currents, could provide more reliable information (see Zhan et al., 2014 and many more). Finally, the schematic circulation, presented in figure 7, based on the annual geostrophic currents is not convincing (at least compared to the black arrows shown in the figure). A seasonal schematic could be more appropriate.

---

## Referee Comment (RC2) · P. L'Hegaret (Referee) · 9 Nov 2018

**General Comment :**

In this article the authors use data from Jason-2 to extend SLA observations from AVISO towards the coasts of the Red Sea. Altimetric products from AVISO are commonly used to describe the open ocean dynamics but their resolutions are coarse near the coasts. The combined satellite dataset is validated with three tide gauges situated along the western coast of the Red Sea and with geostrophic surface velocities estimated from CTD. This new merged satellite product shows good agreement with the other available dataset and allows the authors to have better observations of the SLA

along the coasts.

Once validation of the products, the authors describe the monthly climatological evolution of the the SLA and surface currents, exhibiting the evolution of mesoscale eddies, in size, position and rotation. A month to month analysis of the surface fields describe the observed eddies and link them to the structure previously studied in the scientific literature.

I think this article is well written, the merged dataset allows us to understand the climatological circulation in the Red Sea, where previous satellite dataset allowed only a partial coverage linked to the geography of the basin. Still it lacks some informations of the dataset used to validate the data and the justifications of some diagnosed.

Nevertheless I felt that the last part of the article did not emphasize the main contribution of this study : the calculation of surface currents and SLA along the coast. As I wrote above, the authors did a good job comparing their results with previous studies, and where they agree, but it would be important in my opinion to add informations on where it provides new informations, particularly along the coasts. The conclusion is a little short, and adding these informations will help wrapping the article nicely.

I suggest accepting after minor corrections.

Specific comments :

Methodology : 1/ The SLA from AVISO gives measurements offshore, while the FSM-SLA method extends these measurements toward the coasts. I wonder how are the discontinuities between dataset removed or smoothed ?

2/ On figure 2 the authors show the correlation between the AVSIO and FSM data, how are they calculated where the AVSIO dataset does not provide measurements (again along the coast) ?

Results : 1/ I suggest separating this section in two part, a first with the validation of the method (down to line 17), and a second with the analysis of the SLA.

OSD
2/ About the CTD : on figure 4 the authors display different part of the Red Sea at different periods comparing AVSIO and the FSM-SLA. What are the justifications for these specific area and periods. I think providing a quantitative analysis would help validating the approach.

3/ The visibility of the geostrophic currents and eddies name of figures 5 and 6 have a low visibility. As they exhibit the main results of the study I suggest remapping them by adding a light opaque filter on the SLA and then adding the arrows and names. The same goes for figure 4 where the arrows are difficult to see.

4/ Figure 7 wrap up the paper with a schematic representation of the currents, but, as the authors state, the monsoons have a strong impact on the Red Sea, particularly on its southern edge. I suggest adding a schematic representation for the winter and summer seasons in order to point out the differences in circulations.

---

## Author Comment (AC2) · 22 Nov 2018

The corrected version of the manuscript is attached as a PDF file.

Please also note the supplement to this comment:
https://www.ocean-sci-discuss.net/os-2018-47/os-2018-47-AC2-supplement.pdf
* * *

---

## Author Comment (AC3) · 22 Nov 2018

*Anonymous Referee on "Estimation of geostrophic current in the Red Sea based on Sea level anomalies derived from extended satellite altimetry data" by Ahmed Mohammed Taqi et al. Referee*

Thank you very much for your interest in the manuscript, and for spending your effort and time in the review, comments, and suggestions, which helped in improving the manuscript. The manuscript was modified based on the  Referee comments. The responses to the comments are described below.

**General Comment:**

In this article the authors use data from Jason-2 to extend SLA observations from AVISO towards the coasts of the Red Sea. Altimetric products from AVISO are commonly used to describe the open ocean dynamics but their resolutions are coarse near the coasts. The combined satellite dataset is validated with three tide gauges situated along the western coast of the Red Sea and with geostrophic surface velocities estimated from CTD. This new merged satellite product shows good agreement with the other available dataset and allows the authors to have better observations of the SLA C1 along the coasts. Once validation of the products, the authors describe the monthly climatological evolution of the the SLA and surface currents, exhibiting the evolution of mesoscale eddies, in size, position and rotation. A month to month analysis of the surface fields describe the observed eddies and link them to the structure previously studied in the scientific literature. I think this article is well written, the merged dataset allows us to understand the climatological circulation in the Red Sea, where previous satellite dataset allowed only a partial coverage linked to the geography of the basin.

**Sub comments:**

Sub-comment: Still it lacks some informations of the dataset used to validate the data and the justifications of some diagnosed.

Reply:  The information about each cruise is added in the manuscript. The 2010 cruise data are used entirely, as suggested by reviewer.

Sub-comment: Nevertheless I felt that the last part of the article did not emphasize the main contribution of this study : the calculation of surface currents and SLA along the coast. As I wrote above, the authors did a good job comparing their results with previous studies, and where they

agree, but it would be important in my opinion to add informations on where it provides new informations, particularly along the coasts.

Reply: The revised manuscript was modified accordingly, wherever was needed.

Sub-comment: The conclusion is a little short, and adding these informations will help wrapping the article nicely.

Reply: The conclusion was modified accordingly.

Comment [1] The SLA from AVISO gives measurements offshore, while the FSMSLA method extends these measurements toward the coasts. I wonder how are the discontinuities between dataset removed or smoothed ?

Reply: The AVISO data was removed near the coast using the polygon. The blank area was replaced by the FSMSLA data with space leaving between the two data set according to the width of the sea either one or two grid cells. This gap was filled using kriging interpolation method to smooth the dataset. See figure below which include two example.

[Figure]

Comment [2] On figure 2 the authors show the correlation between the AVSIO and FSM data, how are they calculated where the AVSIO dataset does not provide measurements (again along the coast)

Reply: FSM data was gridded into 0.25°x0.25° and the correlation was estimated for entire Red Sea area. Since FSM data showed better resolution towards the coast, it has been used instead of AVISO data near the coast.

**Comment from Results:**

Comment [1] I suggest separating this section in two part, a first with the validation of the method (down to line 17), and a second with the analysis of the SLA.

Reply: The revised manuscript is modified accordingly

Comment [2] About the CTD: on figure 4 the authors display different part of the Red Sea a different period comparing AVSIO and the FSM-SLA. What are the justifications for these specific area and periods. I think providing a quantitative analysis would help validating the approach.

Reply: The selection of these areas and periods were based upon the available cruise data. In the revised manuscript, the 2010 cruise dataset was entirely compared with our data, which cover larger area

Comment [3] The visibility of the geostrophic currents and eddies name of figures 5 and 6 have a low visibility. As they exhibit the main results of the study I suggest remapping them by adding a light opaque filter on the SLA and then adding the arrows and names. The same goes for figure 4 where the arrows are difficult to see.

Reply: The visibility of the geostrophic currents and eddies names of figures 5 and 6 arrows and names has been changed.

Comment [4] Figure 7 wrap up the paper with a schematic representation of the currents, but, as the authors state, the monsoons have a strong impact on the Red Sea, particularly on its southern edge. I suggest adding a schematic representation for the winter and summer seasons in order to point out the differences in circulations.

Reply: The annual schematic has been changed to the winter and summer seasons see figure 10, in the revised manuscript.

---

## Author Comment (AC4) · 22 Nov 2018

**Estimation of geostrophic current in the Red Sea based on Sea level anomalies derived from extended satellite altimetry data**

Ahmed M Taqi[a,b], Abdullah M Al-Subhi[a], Mohammed A Alsaafani[a] and Cheriyeri P Abdulla[a]

[a]Department of Marine Physics, King Abdulaziz University, Jeddah, Saudi Arabia;
[b]Department of Marine Physics, Hodeihah University, Hodeihah, Yemen

Correspondence to: Ahmed. M. Taqi (ataqi@stu.kau.edu.sa)

**Abstract**

The geostrophic currents near the coast of the Red Sea has a large gap. Due to this, the sea level anomaly (SLA) data of Jason-2 has been reprocessed and extended towards the coast of the Red Sea and merged with AVISO data at the offshore region. The processing has been applied to build a data grid to achieve best results for the SLA and geostrophic current. The results obtained from the new extended data at the coast are more consistent with the observed data (CTD) and hence geostrophic current calculation. The pattern of SLA distribution and geostrophic currents are divided into two seasons; winter season extends from October to May and summer from June to September. The geostrophic currents along the eastern Red Sea flow toward north and southward along the west coast. This flow is modified with the presence of the cyclonic and anticyclonic eddies, which are more concentrated in the central and northern Red Sea. The results show anticyclonic eddies (AE) on the eastern side of the Red Sea and cyclonic eddies (CE) on the western side during winter. During summer, the (CE) are along the eastern side and (AE) along the western side. In summer, cyclonic eddies are more dominant for the entire Red Sea while in winter both cyclonic and anticyclonic eddies are present. Furthermore, the lifespan of cyclonic eddies are longer than that of anticyclonic eddies. This study is the first of this type in the Red Sea which extend SLA and geostrophic

[revised manuscript text omitted]
 $19.5^{\mathrm{o}} - 25^{\mathrm{o}}$N; two cyclonic eddies at $24^{\mathrm{o}} - 25^{\mathrm{o}}$N, and $20^{\mathrm{o}} - 22^{\mathrm{o}}$N, two anticyclonic eddies at $23^{\mathrm{o}} - 24^{\mathrm{o}}$N, and $19.5^{\mathrm{o}} - 20^{\mathrm{o}}$N. From the previous results, it can be seen several cyclonic and anticyclonic eddies distributed all over the Red Sea and these results match those in modelling
studies (Clifford et al., 1997; Eladawy et al., 2017; Sofianos, 2003, 2002, Yao et al., 2014a)

During June, the flow of the geostrophic currents in the northern part reversed its direction. This
accompanies a formation of large cyclonic eddy extending from $25.5^{\circ}$– $27.5^{\circ}$N occupying the
entire width of the Red Sea. To the south of it, another cyclonic eddy observed between $24^{\circ}$ – $25^{\circ}$N
and an anticyclonic eddy between $23^{\circ}$ – $24^{\circ}$N are also noticed during June with a similar strength
during May. The cyclonic eddy seen between $17^{\circ}$ – $20^{\circ}$N during May, is also seen during this
month with more strength. To the south of it, the flow is towards the Bab el-Mandab following
normal summer pattern. The flow pattern along the coast is similar to results of (Chen et al., 2014)
for winter (January to April). The short-term climatology of geostrophic current in the Red Sea is
dominated by cyclonic and anticyclonic eddies all over the Red Sea, and especially in the central
and northern parts of the sea.

During July-September, the flow of the geostrophic currents structure is similar to that of
June with two cyclonic eddies north of $24.5^{\circ}$N and an anticyclonic eddy between $23^{\circ}$ – $24^{\circ}$N. South
of these eddies, another cyclonic eddy extends to $19^{\circ}$N. Furthermore, south of $19^{\circ}$N, there is an
outflow towards the south all over the width of the Red Sea with narrow inflow along the eastern
coast of the Red Sea. The Fig. 6 also shows an anticyclonic between $18^{\circ}$-$19^{\circ}$N and a cyclonic
between $16^{\circ}$-$17^{\circ}$N during August and September. These results are consistent with the results from
previous studies (Clifford et al., 1997; Eladawy et al., 2017; Sofianos, 2003, 2002, Yao et al.,
2014b).

[revised manuscript text omitted]

---

## Author Comment (AC6) · 22 Dec 2018

the version of the corrected manuscript is uploaded as a pdf file

Please also note the supplement to this comment:
https://www.ocean-sci-discuss.net/os-2018-47/os-2018-47-AC6-supplement.pdf

---

## Author Response (AR1)

**Author response**

The authors thank the Editor of "**Ocean Science**" and the two anonymous referees, for their time and efforts in reviewing the manuscript "**os-2018-47**". The manuscript is revised according to the reviewer suggestions and comments, which were very helpful to improve the manuscript. As part of the review process, additional analysis on eddy parameters and their interannual variability are included to the manuscript. I hope that the revised manuscript addressed all the review comments satisfactorily. Kindly consider the manuscript for possible publication in "Ocean Science".

Please note that we have included our colleague **Dr. Cheriyeri P Abdulla** in the author list, as he potentially contributed in the analysis during the review process and helped in revising the manuscript.

The response to the referee comments and the marked-up version of the manuscript is given below.

- 1. Response to referee comments RC1
- 2. Response to referee comments RC2
- 3. Marked-up version of the manuscript

**Response to referee comments RC1**

Anonymous Referee on "Estimation of geostrophic current in the Red Sea based on Sea level anomalies derived from extended satellite altimetry data" by Ahmed Mohammed Taqi et al.

**Anonymous Referee**

Thank you very much for your interest in the manuscript, and for spending your effort and time in the review, comments, and suggestions, which helped in improving the manuscript. The manuscript was modified based on the Anonymous Referee comments. The responses to the comments are described below.

Comments to the Anonymous Referee

**General comment**

The paper "Estimation of geostrophic current in the Red Sea based on Sea level anomalies derived from extended satellite altimetry data" by Taqi et al. focuses on describing the geostrophic currents and eddy field in the Red Sea based on altimetry data, extended to the coast using a method proposed by same authors (Taqi et al., 2017). The first part consists of a continuation of the validation of the method (adding hydrographic data for estimating the geostrophic velocity) and the second part provides an analysis of the monthly climatology of the sea level anomaly (SLA) and the corresponding surface currents (averaging 6 years satellite data).

Comment [1] The validation part provides very little additional analysis compared to the Taqi et al., 2017, while there is no information and/or reference related to the cruises that were used for estimating the geostrophic currents (lines 107-110). Actually, after checking the reference provided later in the text (e.g. Bower and Farrar, 2015) the cruise(s) covered a much larger area than the one used and shown in this paper. It is not understood why the authors selected the specific regions to perform the validation.

Reply: The information about each cruise is added in the manuscript. The 2010 cruise data are used entirely, as suggested by reviewer.

Comment [2] The cruises also used an LADCP and thus the adoption of 700 m reference level seems arbitrary (actually most of the stations are shallower than that).

Reply: The stations have depths that very from 150 up to 1800 m. However, most of the stations exceed the 500 m depth, accordingly the level of on motion se to 500m. between deeper and shallow, and we returned the calculations for average depth at a reference level of 500 meters. it was mean deep in stations above 500 m.

Comment [3] The comparison and error estimation is very qualitative (comparing figures) and in figure 4a&b (the largest area covered) it is impossible to visualize the results.

Reply: As suggested a quantitative analysis is done for the data and added the same in the revised manuscript from line 186 to 191.

Comment [4] The second part is very weak, merely describing the twelve monthly SLA/geostrophic velocity figures. The methodology of averaging 6 years of SLA data to describe the climatology of the complex Red Sea eddy field is not appropriate. While the basin-scale seasonal variability of the SLA can benefit very little from the new method of extending the data to the coast (this comparison is not shown), the averaging could mask the eddy field and produce artificial features. More advanced methods, including the interannual variability of the SLA/geostrophic currents, could provide more reliable information (see Zhan et al., 2014 and many more).

Reply: A) We agree with the reviewer that the averaging of 6 years data will not give the variable eddies in the Red Sea, even it shows the permanent eddies clearly. Please see the attached figure, which compares the climatology with SLA of 2010. The patterns were similar, with small differences. The main differences are the short timing eddies are not visible in the climatology, but the general features of variability of circulation is present.

b) As suggested by the reviewer, more analysis on the SLA/geostrophic currents and the statistical analysis of eddies in the Red Sea are added in the manuscript from line 296 to 318.

Comment [5] Finally, the schematic circulation, presented in figure 7, based on the annual geostrophic currents is not convincing (at least compared to the black arrows shown in the figure). A seasonal schematic could be more appropriate.

---

## Referee Report (RR1)

**Estimation of geostrophic current in the Red Sea based on Sea level anomalies derived from extended satellite altimetry data**

**Second review**
* * *
**General comment**

In this article, the authors use altimetric data extended toward the coast to describe the monthly climatological circulation in the Red Sea. This study provides a spatial configuration of the mesoscale eddies in the basin and their evolution along a climatological year. The authors also discuss the interannual variability of this surface circulation.

After taking into account the different reviews, the overall quality of the manuscript has improved. Particularly the data and method and the validation sections, where the authors provided more informations on their calculations as well as comparisons with in-situ observations.

The structure of the paper and its aim is clear, but in my opinion some figures remain unclear and the text still contains some grammatical errors. I recommend publication after minor revisions.
* * *
**Specific comments**

**FIGURES**
Figure 4 : Change the last colorbar to have matching fonts. I also suggest changing the maximal and minimal value of the colobar as figures 4c to 4f only exhibit one color.

Figure 5 and 6 : I still think these figures are not readable which is a shame as they show the main result of the paper. For each month, only a color is exhibited, arrows are quite small, and for the months of June through September, the « AE » and « CE » fonts are not readable over a yellow color. As the SLA can be centered at 0, a « balance colorbar » (as the one found here : https://matplotlib.org/cmocean/) could help improve greatly the figure.

Figure 7 : Same remark as the previous figure for the axis of the colorbars.

I provide hereafter some grammatical corrections for the english. But as non-native english speaker I do not think it covers everything in the paper. I suggest having a thorough review of the english in the text.

**ABSTRACT**
Line 19 : I suggest changing the end of the sentence by « except along the eastern coast where they flow in the opposite direction».

Line 21 : « except a small part near the eastern coast ».

Line 21 : « This flow is modified by the presence of ».

**INTRODUCTION**
Line 31 : « between the continents ».

Line 32 : te words « latitude » and « longitude » are not necessary.

Line 51 : « that are often present ».

Line 54 : « 18°N and 24°N »

Line 70 : « from the coastline ».

Line 84 I suggest « advect » instead of « circulate ».

**RESULTS AND DISCUSSION**
Line 159 : « on the other hands » should not be used. Change to something like « Oppositely, near the coasts, weak correlation coefficient are found between the two datasets ».

Line 175 : « shows ».

Line 199 : « three cruises ».

Line 224 : « starts ».

**CONCLUSION**
Line 329 to 331 : This sentence should be removed as it is repeated in the last paragraph.

Line 332 : « shows ».

Line 334 : « Geostrophic current ».

Line 335-336 : « except along the astern coast where they flow in the opposite direction ».

Line 337 : « except a small part […] and of the western coast ».

Line 341 : « northern Red Sea ».

Line 341-343 : Please rewrite this sentence.

---

## Author Response (AR2)

**Author response**

The authors thank the Editor of "**Ocean Science**" and the two anonymous referees, for their time and efforts in reviewing the manuscript "**os-2018-47**". The manuscript is revised according to the reviewer suggestions and comments, which were very helpful to improve the manuscript. As part of the review process, additional analysis on eddy parameters and their interannual variability are included to the manuscript. I hope that the revised manuscript addressed all the review comments satisfactorily. Kindly consider the manuscript for possible publication in "Ocean Science".

The response to the referee comments and the marked-up version of the manuscript is given below.

1. Editor comments EC
2. Response to referee comments RC1
3. Response to referee comments RC2
4. Marked-up version of the manuscript

*Response to Editor comments on "Estimation of geostrophic current in the Red Sea based on Sea level anomalies derived from extended satellite altimetry data" by Ahmed Mohammed Taqi et al.*

Thank you very much for your interest in the manuscript, and for spending your effort and time in the review, comments, and suggestions, which helped in improving the manuscript. The manuscript was modified based on the comments. The responses to the comments are described below.

**Comment [1]** Lines 127-128. I think you should include here some of your response to referee 2 comment 1 about removing discontinuity between AVISO and FSMSLA. Maybe refer to Taqi et al (2017) for more detail?

Reply: modified in the manuscript.

Line number: [130-133].

**Comment [2]** Lines 131, 135. Two equations (2)!

Reply: modified in the manuscript.

Line number: [136].

**Comment [3]** Lines 137-138. The choice of level of no motion is still arbitrary. You have shown only pragmatism and not logic in choosing 500 m. There might be flow at all depths or indeed significant flow only in the top 200 m. Evidence should be used in the choice and Bower and Farrar (2015) refer to ship-borne ADCP measurements usually down to 600 m which would be good evidence. Does the choice between 500m and 700m make a significant difference to the near-surface geostrophic currents?

Reply: This part is also clarified in the reply to referee 1 comment 2. The following paragraph is added to the manuscript.

Previous study by Quadfasel and Baudner (1993) used 400 m as level of no motion to calculate geostrophic current in the Red Sea. Based on ADCP measurements, Bower and Farrar (2015) shown that, on average, 75–95 % of the vertical shear is occurred over the top 200 m of the water column. Moreover, the ADCP measurements of current speed below 500 m is very small; about

~0.06m/s at 600 m depth (Bower and Farrar, 2015). Therefore, expecting negligible variability below 500 m, a depth of 500 m was selected as a level of no motion. We have compared the geostrophic current corresponding to level of no motion at 500m and 700m. The observed difference between both are negligibly small.

Line number: [142-150].

**Comment [4]** Figure 2. You have not answered the Referee 2 question about how you show correlation with AVISO data near the coast where there are no AVISO data.

Reply: The answer to referee 2 question is improved for better understanding as follows.

Please note that the Gridded AVISO data is available in the coastal region as well as in the offshore area, but the accuracy of the AVISO data near the coast is questionable, especially in narrow basin like the Red Sea. FSM data was gridded to the same resolution as AVISO (0.25°x0.25°). We compared the data along the coast from both data products (AVISO and FSM) against the in-situ (tide gauge) measurements. The statistics of the comparison is shown in Table 2. The FSM data showed better correlation against the in-situ data in all the cases. For this reason, we have used FSM instead of AVISO data near the coast.

**Comment [5]** Figure 4. Referee 2 asked for justification for these specific areas and periods. Your revised text explains the times but not the areas; Referee 1 found that the cruise in Bower and Farrar (2015) covered a larger area. This is another case where the referee suggests more quantitative analysis.

Reply: The reply to referee 2 comment #1 is modified as follows based on your comment.

To validate the geostrophic current, we used the available in situ profiles during the available periods and regions. The in situ data include the following cruises; 1) March 16 to 29, 2010 onboard R/V Aegaeo between 22°N to 28°N along the eastern Red Sea, 2) April 3 to 7, 2011 onboard Poseidon between 17°N to 22°N in the central eastern Red Sea, and 3) October 16 to 19, 2011 onboard the same vessel between 19°N to 23°N in the central eastern Red Sea,. We understand that we are limited in space and time because of the spatial and temporal distribution of the available cruises. Regarding the cruise 1 by Bower and Farrar (2015), unfortunately, we do not have complete data set used by the Bower and Farrar (2015), and therefore we used only the available profiles for validation.

The text in the manuscript is modified accordingly.

Line number: [114-122].

Regarding the quantitative analysis please refer to reply to referee 1 comment #3, where the manuscript modified accordingly.

 Line number: [201-206, and Table 3].

**Comment [6]** Table 3 for March 2010. I am wondering if the values are all correct in that the RMSE and Stdv are both worse for FSM-SLA than for AVISO but the correlation coefficient is better. I can understand that the bias might affect RMSE but it should not affect the Stdv? There is a question as to whether Table 3 is informative in that agreement with CTD-derived geostrophic current is not necessarily agreement with the true current.

Reply: modified the table 3 in the manuscript.

We verified the values in table 3 and found that the values corresponding to the FSM-SLA during March-2010 is not correct. We revised all MATLAB codes used for calculation and we discovered an error in one of the codes. For this we have modified the code and recalculated the statistical parameters and replaced in the new table3. Please note that, the FSM-SLA is less biased with lower RMSE value and better correlation in all cases.

**Comment [7]** Line 196. Referee 1 questioned the 6-year averaging. You refer to 6 years but the SLA data only span part of 2009 and 2014. Then figure 8 for "2014" will contain some bias opposite to any October-December seasonal signal?

Reply: The manuscript is modified.

We agree that the previous data were from June 2009 to October 2014. We extend the data to the end of 2014 following (Taqi et al.2017). We processed data for all available tracks for the period from October (cycle 232) to December (cycle 239) following the same process as the previous data. After the extension of the data, figure 8 has been updated accordingly. Regarding referee 1 question, please refer to reply to comment # 4 of referee 1.

Line number: [101].

**Comment [8]** Page 13, figure 5. Where is the arrow showing the "reference current length =0.5m/s"?

Reply: modified in figure 5 and add reference current length.

**Comment [9]** Lines 274-275. "amplitude of eddies in the Red Sea is about 4 cm which is an agreement with global values". Is there any reason to hope for or expect "agreement". Lower latitudes and relatively narrow Red Sea might lead one to expect smaller eddy amplitudes.

Reply: The manuscript is modified, and the following paragraph is included.

The amplitude of an eddy has been defined as the differences between the estimated basic height of the eddy boundary and the extremum value of SLA inside the eddy interior parts. The mean amplitude for anticyclonic is between 1.3 cm at southern Red Sea to 5.3 cm at northern Red Sea and for cyclonic eddy is between 1.6 cm at southern Red Sea to 4.2 cm at northern Red Sea. The result indicates the average value of eddy amplitude in the Red Sea (including low latitude and high latitude) is about 4 cm, which is within the reasonable range defined by (Chelton et al., 2011).

Line number: [284-290].

**Comment [10]** Figure 8. Earlier Reviewer 1 commented "The comparison and error estimation is very qualitative (comparing figures) and in figure 4a&b (the largest area covered) it is impossible to visualize the results." I think the same now applies to the figures in response to Referee 1 comment [4] and to new figure 8. Perhaps the revision should include some statistics about the difference between 2010 and climatology. Or preferably, what is the standard deviation of SLA in the panels of figure 8 compared with the standard deviation of the "6-year mean"?

Reply:

Regarding referee comment #3, a statistical analysis to compare the FSM-SLA and AVISO derived geostrophic current with that derived from CTD, were added to the manuscript text and summarized in Table 3.

Line number: [201-206].

Regarding Figure 8 and referee comment # 4, We have computed the statistics, which is summarized in the manuscript (Table 4).

Line number: [308-316].

**Comment [11]** Figure 8 arrows. Do these show direction only (they seem to all have the same length)?

Reply: modified in the figure 8.

**Comment [12]** Referee 2 comment "it would be important in my opinion to add informations on where it provides new informations, particularly along the coasts." You have many statements about geostrophic flows, including along the coast, and about cyclonic and anticyclonic eddies. However, you also have many statements that what you find agrees with previous findings. I think the referee wants you to say what you have found that was NOT previously found. I want this too as an editor of Ocean Science which is for publishing new knowledge and understanding.

Reply: The manuscript is modified, and the following text is included in the "conclusion" section.

The finding of this paper considered the first of its type in the Red Sea for extending SLA and geostrophic currents to the coast beside giving more details of eddies spatial and temporal variabilities in the coastal region. In addition, in winter, the cyclonic eddies are along the west coast and anticyclonic eddies on the east side of the Red Sea, while in summer it is the opposite. Also, in some locations there is a noticeable change from anticyclonic during winter to cyclonic during summer and vice versa between 26.3°N –27.5°N.

The major new findings from the present study include the monthly geostrophic pattern in the Red Sea which has not been published before. Seasonally, the geostrophic currents in summer are flowing northward except the eastern coast which flows in the opposite. In winter, currents flow to the north for the entire sea except small part of the eastern coast (22°N-24°N) and the western coast (23°N-20°N).

Line number: [334-338, 342-344, 351-353].

**Response to referee comments RC1**

*Anonymous Referee on "Estimation of geostrophic current in the Red Sea based on Sea level anomalies derived from extended satellite altimetry data" by Ahmed Mohammed Taqi et al.*

*Anonymous Referee*

Thank you very much for your interest in the manuscript, and for spending your effort and time in the review, comments, and suggestions, which helped in improving the manuscript. The manuscript was modified based on the Anonymous Referee comments. The responses to the comments are described below.

Comments to the *Anonymous Referee*

**General comment**

The paper "Estimation of geostrophic current in the Red Sea based on Sea level anomalies derived from extended satellite altimetry data" by Taqi et al. focuses on describing the geostrophic currents and eddy field in the Red Sea based on altimetry data, extended to the coast using a method proposed by same authors (Taqi et al., 2017). The first part consists of a continuation of the validation of the method (adding hydrographic data for estimating the geostrophic velocity) and the second part provides an analysis of the monthly climatology of the sea level anomaly (SLA) and the corresponding surface currents (averaging 6 years satellite data).

**Comment [1]:**The validation part provides very little additional analysis compared to the Taqi et al., 2017, while there is no information and/or reference related to the cruises that were used for estimating the geostrophic currents (lines 107-110). Actually, after checking the reference provided later in the text (e.g. Bower and Farrar, 2015) the cruise(s) covered a much larger area than the one used and shown in this paper. It is not understood why the authors selected the specific regions to perform the validation.

**Reply**: To validate the geostrophic current, we used the available in situ profiles during the available periods and regions. The in situ data include the following cruises; 1) March 16 to 29, 2010 onboard R/V Aegaeo between 22°N to 28°N along the eastern Red Sea, 2) April 3 to 7, 2011 onboard Poseidon between 17°N to 22°N in the central eastern Red Sea, and 3) October 16

to 19, 2011 onboard the same vessel between 19°N to 23°N in the central eastern Red Sea,. We understand that we are limited in space and time because of the spatial and temporal distribution of the available cruises. Regarding the cruise 1 by Bower and Farrar (2015), unfortunately, we do not have complete data set used by the Bower and Farrar (2015), and therefore we used only the available profiles for validation.

The text in the manuscript is modified accordingly.

Line number: [114-122].

**Comment [2]:** The cruises also used an LADCP and thus the adoption of 700 m reference level seems arbitrary (actually most of the stations are shallower than that).

**Reply**: The manuscript is modified, and the following paragraph is included .

Previous study by Quadfasel and Baudner (1993) used 400 m as level of no motion to calculate geostrophic current in the Red Sea. Based on ADCP measurements, Bower and Farrar (2015) shown that, on average, 75–95 % of the vertical shear is occurred over the top 200 m of the water column. Moreover, the ADCP measurements of current speed below 500 m is very small; about ~0.06m/s at 600 m depth (Bower and Farrar, 2015). Therefore, expecting negligible variability below 500 m, a depth of 500 m was selected as a level of no motion. We have compared the geostrophic current corresponding to level of no motion at 500m and 700m. The observed difference between both are negligibly small.

Line number: [142-150].

**Comment [3]:** The comparison and error estimation is very qualitative (comparing figures) and in figure 4a&b (the largest area covered) it is impossible to visualize the results.

**Reply**: As suggested a quantitative analysis is done for the data and added the same in the revised manuscript.

Line number: [201-206, and Table 3].

**Comment [4]:** The second part is very weak, merely describing the twelve monthly SLA/geostrophic velocity figures. The methodology of averaging 6 years of SLA data to describe the climatology of the complex Red Sea eddy field is not appropriate. While the basin-scale seasonal variability of the SLA can benefit very little from the new method of extending

the data to the coast (this comparison is not shown), the averaging could mask the eddy field and produce artificial features. More advanced methods, including the interannual variability of the SLA/geostrophic currents, could provide more reliable information (see Zhan et al., 2014 and many more).

**Reply**: A) We agree with the reviewer that the averaging of 6 years data will not give the variable eddies in the Red Sea, even it shows the permanent eddies clearly. Please see the attached figure, which compares the climatology with SLA of 2010. The patterns were similar, with small differences. The main differences are the short timing eddies are not visible in the climatology, but the general features of variability of circulation is present.

b) As suggested by the reviewer, more analysis on the SLA/geostrophic currents and the statistical analysis of eddies in the Red Sea are added in the manuscript in fig.7 .
Line number: [278-297].
We have also studied the interannual variability in the SLA and geostrophic current in Fig. 8 and Table 4
Line number: [308-316].

**Comment [5]:** Finally, the schematic circulation, presented in figure 7, based on the annual geostrophic currents is not convincing (at least compared to the black arrows shown in the figure). A seasonal schematic could be more appropriate.

**Reply**: The annual schematic has been changed to the winter and summer seasons see figure 10, in the revised manuscript.

[Figure]

**Figure compared between year 2010 and monthly climatology for geostrophic current and Sea level anomaly (Reference current length =0.5 m/sec)**

[Figure]

Figure 6 As figure 5 for July to December

*Anonymous Referee on "Estimation of geostrophic current in the Red Sea based on Sea level anomalies derived from extended satellite altimetry data" by Ahmed Mohammed Taqi et al.*
*Referee*

Thank you very much for your interest in the manuscript, and for spending your effort and time in the review, comments, and suggestions, which helped in improving the manuscript. The manuscript was modified based on the Referee comments. The responses to the comments are described below.

*Comments*

**General Comment:**

In this article the authors use data from Jason-2 to extend SLA observations from AVISO towards the coasts of the Red Sea. Altimetric products from AVISO are commonly used to describe the open ocean dynamics but their resolutions are coarse near the coasts. The combined satellite dataset is validated with three tide gauges situated along the western coast of the Red Sea and with geostrophic surface velocities estimated from CTD. This new merged satellite product shows good agreement with the other available dataset and allows the authors to have better observations of the SLA C1 along the coasts. Once validation of the products, the authors describe the monthly climatological evolution of the the SLA and surface currents, exhibiting the evolution of mesoscale eddies, in size, position and rotation. A month to month analysis of the surface fields describe the observed eddies and link them to the structure previously studied in the scientific literature. I think this article is well written, the merged dataset allows us to understand the climatological circulation in the Red Sea, where previous satellite dataset allowed only a partial coverage linked to the geography of the basin.

**Comment [1]**: Still it lacks some informations of the dataset used to validate the data and the justifications of some diagnosed.

Reply: To validate the geostrophic current, we used the available in situ profiles during the available periods and regions. The in situ data include the following cruises; 1) March 16 to 29, 2010 onboard R/V Aegaeo between 22°N to 28°N along the eastern Red Sea, 2) April 3 to 7, 2011 onboard Poseidon between 17°N to 22°N in the central eastern Red Sea, and 3) October 16 to 19, 2011 onboard the same vessel between 19°N to 23°N in the central eastern Red Sea,. We understand that we are limited in space and time because of the spatial and temporal distribution of the available cruises. Regarding the cruise 1 by Bower and Farrar (2015), unfortunately, we do not have complete data set used by the Bower and Farrar (2015), and therefore we used only the available profiles for validation.

The text in the manuscript is modified accordingly.

Line number: [114-122].

**Comment [2]**: Nevertheless I felt that the last part of the article did not emphasize the main contribution of this study : the calculation of surface currents and SLA along the coast. As I wrote above, the authors did a good job comparing their results with previous studies, and where they agree, but it would be important in my opinion to add informations on where it provides new informations, particularly along the coasts.

Reply: The manuscript is modified, and the following text is included in the "conclusion" section.

The finding of this paper considered the first of its type in the Red Sea for extending SLA and geostrophic currents to the coast beside giving more details of eddies spatial and temporal variabilities in the coastal region. In addition, in winter, the cyclonic eddies are along the west coast and anticyclonic eddies on the east side of the Red Sea, while in summer it is the opposite. Also, in some locations there is a noticeable change from anticyclonic during winter to cyclonic during summer and vice versa between 26.3°N –27.5°N.

The major new findings from the present study include the monthly geostrophic pattern in the Red Sea which has not been published before. Seasonally, the geostrophic currents in summer are flowing northward except the eastern coast which flows in the opposite. In winter, currents flow to the north for the entire sea except small part of the eastern coast (22°N-24°N) and the western coast (23°N-20°N).

Line number: [334-338, 342-344, 351-353].

**Comment[3]**: The conclusion is a little short, and adding these informations will help wrapping the article nicely.

Reply: The conclusion in the manuscript is improved.

**Comment [4]:** The SLA from AVISO gives measurements offshore, while the FSMSLA method extends these measurements toward the coasts. I wonder how are the discontinuities between dataset removed or smoothed.

**Reply**: The AVISO data was removed near the coast using the polygon.  The blank area was replaced by the FSMSLA data with space leaving between the two data set according to the width of the sea either one or two grid cells. This gap was filled using kriging interpolation method to smooth the dataset. See figure below which include two examples.

[Figure]

**Comment [5]:** On figure 2 the authors show the correlation between the AVSIO and FSM data, how are they calculated where the AVSIO dataset does not provide measurements (again along the coast)

**Reply**: Please note that the Gridded AVISO data is available in the coastal region as well as in the offshore area, but the accuracy of the AVISO data near the coast is questionable,

especially in narrow basin like the Red Sea. FSM data was gridded to the same resolution as AVISO (0.25°x0.25°). We compared the data along the coast from both data products (AVISO and FSM) against the in-situ (tide gauge) measurements. The statistics of the comparison is shown in Table 2. The FSM data showed better correlation against the in-situ data in all the cases. For this reason, we have used FSM instead of AVISO data near the coast.

**Comment from Results:**

**Comment [6]:** I suggest separating this section in two part, a first with the validation of the method (down to line 17), and a second with the analysis of the SLA.

**Reply**: The revised manuscript is modified accordingly.

**Comment [7]** About the CTD: on figure 4 the authors display different part of the Red Sea a different period comparing AVSIO and the FSM-SLA. What are the justifications for these specific area and periods. I think providing a quantitative analysis would help validating the approach.

**Reply**: To validate the geostrophic current, we used the available in situ profiles during the available periods and regions. The in situ data include the following cruises; 1) March 16 to 29, 2010 onboard R/V Aegaeo between 22°N to 28°N along the eastern Red Sea, 2) April 3 to 7, 2011 onboard Poseidon between 17°N to 22°N in the central eastern Red Sea, and 3) October 16 to 19, 2011 onboard the same vessel between 19°N to 23°N in the central eastern Red Sea,. We understand that we are limited in space and time because of the spatial and temporal distribution of the available cruises. Regarding the cruise 1 by Bower and Farrar (2015), unfortunately, we do not have complete data set used by the Bower and Farrar (2015), and therefore we used only the available profiles for validation.

The text in the manuscript is modified accordingly.

Line number: [114-122].

**Comment [8]** The visibility of the geostrophic currents and eddies name of figures 5 and 6 have a low visibility. As they exhibit the main results of the study I suggest remapping them by adding a light opaque filter on the SLA and then adding the arrows and names. The same goes for figure 4 where the arrows are difficult to see.

**Reply**: The visibility of the geostrophic currents and eddies names of figures 5 and 6 arrows and names has been changed.

**Comment [9]** Figure 7 wrap up the paper with a schematic representation of the currents, but, as the authors state, the monsoons have a strong impact on the Red Sea, particularly on its southern edge. I suggest adding a schematic representation for the winter and summer seasons in order to point out the differences in circulations.

**Reply**: The annual schematic has been changed to the winter and summer seasons see figure 10, in the revised manuscript.

[revised manuscript text omitted]

---

## Author Response (AR3)

To:

Prof. John M. Huthnance,

Topic Edito

Ocean Science.

Sub:- Submission of revised manuscript [# os-2018-47].

Dear Prof. John M. Huthnance,

Thank you for your email dated on .01th March 2019 with the reviews of our manuscript titled "
Estimation of geostrophic current in the Red Sea based on Sea level anomalies derived from
extended satellite altimetry data".

We have revised the manuscript in accordance with the comments from the Referee 2 and Topic Editor.
The comments really helped us in improving the manuscript further. We thank you and the reviewers for
the constructive criticism.

A point-by-point response to the comments of the reviewer is enclosed along with the revised manuscript.

We look forward to hearing from you.

Thanking you,

Yours sincerely,

(A. M. Taqi)

**Responses to the Referee comments:**

General comment.

Thank-you again for your revised manuscript. In the following, I copy "Referee 2 comments" (which you may have seen) but have added in some of my own and some which attempt to represent the original concerns of Referee 1 which still need more attention in my opinion. (Unfortunately, I do not have a re-review from Referee 1).

"In this article, the authors use altimetric data extended toward the coast to describe the monthly climatological circulation in the Red Sea. This study provides a spatial configuration of the mesoscale eddies in the basin and their evolution along a climatological year. The authors also discuss the interannual variability of this surface circulation.

After taking into account the different reviews, the overall quality of the manuscript has improved. Particularly the data and method and the validation sections, where the authors provided more informations on their calculations as well as comparisons with in-situ observations.

The structure of the paper and its aim is clear, but in my opinion some figures remain unclear and the text still contains some grammatical errors. I recommend publication after minor revisions.

Response:

We thank the topic editor and Referee for their construction comments. Following is the response to specific comments point by point.

**Specific comments**

FIGURES

**Comments [1]:** Figure 4, Change the last colorbar to have matching fonts. I also suggest changing the maximal and minimal value of the colobar as figures 4c to 4f only exhibit one color.

**Reply:** The colorbar is changed as suggests.

**Comments [2]:** Figure 5 and 6: I still think these figures are not readable which is a shame as they show the main result of the paper. For each month, only a color is exhibited, arrows are quite small, and for the months of June through September, the «

AE » and « CE » fonts are not readable over a yellow color. As the SLA can be centered at 0, a « balance colorbar » (as the one found here:

https://matplotlib.org/cmocean/) could help improve greatly the figure." I think March and April are the wrong way around (either in layout or labelling).

**Reply:** Figure 5 and 6 are modified for more clarity with appropriate colorbar and clear arrows of suitable sizes. The labeling also has been modified.

**Comments [3]:** "Figure 7: Same remark as the previous figure for the axis of the colorbars." This could be outside the panel border if that helps.

**Reply:** Figure 7 modified accordingly unified and only one colorbar presented for all panels.

**Comments [4]:** "I provide hereafter some grammatical corrections for the english. But as non-native English speaker I do not think it covers everything in the paper. I suggest having a thorough review of the English in the text.

**Reply:** The manuscript is carefully reviewed for English after modifying for those comments.

*ABSTRACT"*

**Comments [5]:** Line 10: delete initial "The".

**Reply:** The manuscript is modified.  Line number: [10].

**Comments [6]:** Lines 10-11. "Due to this," -> "Hence"

**Reply:** The manuscript is modified. Line number: [10].

**Comments [7]:** Line 13. ". . data set . ."

**Reply:** The manuscript is modified. Line number: [13].

**Comments [8]:** Line 16. "patterns"

**Reply:** The manuscript is modified. Line number: [16].

**Comments [9]:** "Line 19 : I suggest changing the end of the sentence by « except along the eastern coast where they flow in the opposite direction».

**Reply:** The manuscript is modified. The entire paragraph from line 18-23 is modified. Line number: [18-23].

**Comments [10]:** Line 21: « except for a small area near the eastern coast ».

**Reply:** The manuscript is modified. Line number: [20-22].

**Comments [11]:** Line 21: « This flow is modified by the presence of »."

**Reply:** The manuscript is modified.  Line number: [20-22].

**Comments [12]:** Line 26. Omit "while in winter both cyclonic and anticyclonic eddies are present" (unnecessary repetition)

**Reply:** The manuscript is modified. Line number: [26].

*"INTRODUCTION"*

**Comments [13]:** Line 31: « between the continents ».

**Reply:** The manuscript is modified. Line number: [31].

**Comments [14]:** Line 32: the words « latitude » and « longitude » are not necessary.

**Reply:** The manuscript is modified. Line number: [32].

**Comments [15]:** Line 52: « that are often present ».

**Reply:** The manuscript is modified. Line number: [52].

**Comments [16]:** Line 54: « 18°N and 24°N »

**Reply:** The manuscript is modified. Line number: [54].

**Comments [17]:** Line 70: « from the coastline ».

**Reply:** The manuscript is modified. Line number: [70].

**Comments [18]:** Line 84: I suggest « advect » instead of « circulate ».”

**Reply:** The manuscript is modified. Line number: [84].

**Comments [19]:** Line 90. Delete "coastal region where in the"

**Reply:** The manuscript is modified.  Line number: [89].

*Section 2.*

**Comments [20]:** Line 132. ". . sea: either . ."

**Reply:** The manuscript is modified. Line number: [131].

**Comments [21]:** Line 142. "very" -> "vary"

**Reply:** The manuscript is modified. Line number: [141].

**Comments [22]:** Line 145. Delete "is"

**Reply:** The manuscript is modified. Line number: [144].

**Comments [23]:** Line 150. "negligibly small". To answer referee 1, I think you should give a value (of the RMS difference, and compare that with the RMS value of the surface current).

**Reply:** The observed difference between the surface geostrophic current corresponding to level of no motion at 500m and 700m are negligibly small, with RMSE around 0.003 m/s at surface.

Line number: [149].

*"RESULTS AND DISCUSSION*

**Comments [24]:** Line 159 : « on the other hands » should not be used. Change to something like « In contrast, near the coasts, weak correlation is found between the two datasets, the correlation coefficient being 0.45 to 0.7 ».”

**Reply:** The manuscript is modified. Line number: [158-159].

**Comments [25]:** Line 165. Better ". . observed coastal tide gauge data . ."

**Reply:** The manuscript is modified. Line number: [164].

**Comments [26]:** Lines 174-176. This sentence repeats what is in previous text and Table 2.

**Reply:** The sentence is removed from manuscript.

**Comments [27]:** "Line 175 : « shows »."

**Reply:** The manuscript is modified. Line number: [174].

**Comments [28]:** Line 183. "central" – why this restriction?

**Reply:** The region was selected based on the availability and distribution of cruise data.

**Comments [29]:** Line 189. "in agreement with Bower and Farrar (2015) findings" Is this now for all their data (referee 1 comment)?

**Reply:** Yes, it agrees with Bower and Farrar (2015) during October 2011 cruise.

**Comments [30]:** "Line 199 : « three cruises »."

**Reply:** The manuscript is modified. Line number: [198].

**Comments [31]:** Table 3 and Table 4. What is the meaning of Standard Deviation as distinct from RMSE here? Why of all the values given is Stdv > RMSE only for the top row of Table 3 – is there a mistake here?

**Reply:** Estimation RMSE and STD were rechecked and Tables 3 and 4 are modified accordingly. From the row calculations it's clear that RMSE and STD give the same test, so we kept only RMSE in Table 3 since we compare the model and observed values.

**Table 3 After check**

| After check | Month-Year | Bias (m/s) | RMSE (m/s) | σ (m/s) | CC | P-Value |
|---|---|---|---|---|---|---|
| FSM-SLA | Mar-2010 | -0.0085 | 0.065 | 0.065 | 0.54 | 0.01 |
| AVISO | | -0.01 | 0.08 | 0.08 | 0.48 | 0.14 |
| FSM-SLA | Apr-2011 | -0.28 | 0.31 | 0.31 | 0.61 | 0.02 |
| AVISO | | -0.87 | 0.89 | 0.89 | 0.44 | 0.13 |
| FSM-SLA | Oct-2011 | -0.19 | 0.49 | 0.49 | 0.53 | 0.10 |
| AVISO | | -0.51 | 0.70 | 0.70 | 0.49 | 0.16 |

Final Table 3 (which is presented in the manuscript)

| | Month-Year | Bias (m/s) | RMSE (m/s) | CC | P-Value |
|---|---|---|---|---|---|
| FSM-SLA | Mar-2010 | -0.0085 | 0.065 | 0.54 | 0.01 |
| AVISO | | -0.01 | 0.08 | 0.48 | 0.14 |
| FSM-SLA | Apr-2011 | -0.28 | 0.31 | 0.61 | 0.02 |
| AVISO | | -0.87 | 0.89 | 0.44 | 0.13 |
| FSM-SLA | Oct-2011 | -0.19 | 0.49 | 0.53 | 0.10 |
| AVISO | | -0.51 | 0.70 | 0.49 | 0.16 |

While in Table 4 we kept only STD since we compare the climatology and annual mean

**Table 4 After check**

| Year | bias | RMSE | σ | CC |
|---|---|---|---|---|
| 2010 | -0.01 | 0.03 | 0.03 | 0.54 |
| 2011 | -0.01 | 0.02 | 0.02 | 0.77 |
| 2012 | -0.01 | 0.02 | 0.02 | 0.55 |
| 2013 | -0.01 | 0.03 | 0.03 | 0.79 |
| 2014 | -0.02 | 0.05 | 0.05 | 0.73 |
| Note: The p-value corresponding to all comparison is very low (P<0.0001), indicating that the results from correlation are significant. | | | | |

Final Table 4 (which is presented in the manuscript)

| Year | bias | σ | CC |
|---|---|---|---|
| 2010 | -0.01 | 0.03 | 0.54 |
| 2011 | -0.01 | 0.02 | 0.77 |
| 2012 | -0.01 | 0.02 | 0.55 |
| 2013 | -0.01 | 0.03 | 0.79 |
| 2014 | -0.02 | 0.05 | 0.73 |
| Note: The p-value corresponding to all comparison is very low (P<0.0001), indicating that the results from correlation are significant. | | | |

**Comments [32]:** "Line 224 : « starts »."

**Reply:** The manuscript is modified. Line number: [223].

**Comments [33]:** Section 3.2. Figures 5, 6 show monthly patterns from averaging the 5- or 6-years month by month (as I understand it). Reviewer 1 questioned this. Differences between these monthly figures are only significant if they are greater than the differences between all the Januarys, all the Februarys etc. So I think you also need to show maps of the standard deviations of the January maps, February maps etc., i.e. 12 maps corresponding to Figures 5, 6 which show a measure of the uncertainty of Figures 5, 6.

**Reply:** Please refer to the figure blow, which show the maps of the standard deviations of all months which reveals low standard deviations.

[Figure]

**Comments [34]:** Line 264. Better ". . October but an anticyclonic eddy forms between . .

**Reply:** The manuscript is modified. Line number: [263].

**Comments [35]:** Lines 305-307 is very unclear. Why should the annual and 6-year mean values be the same? How can all the biases be negative if the mean is of all the contributing years? "less" than what?

**Reply:** Please refer to reply to comment (31), where the RMSE is reposed from this Table.

The text is modified of more clarity as follow.

"The statistical analysis between annual FSM-SLA with 6-year climatology shown in Table 4. The correlation is significant for all the years with standard deviation less than 0.1. the bias is very small regardless its sign."

**Comments [36]:** Lines 316-317. ". . along the middle from 20°N to the north." seems to partly repeat and partly contradict "The mean flow is toward the north mostly along the western coast and center up to 23°N . . ."

**Reply:** The manuscript is modified as follow.

"During winter, the mean flow is toward the north all over the entire width of Red Sea, this result agrees with Sofianos and Johns, (2003).  In addition, a southward flow of geostrophic currents were observed along the eastern coast at (22°N-24°N) and the western coast at (23°N-20°N). During summer, the flow is towards the south along the western side of the sea while in southern part the flow spreads across most of the width of the Red Sea with a narrow northward flow near the eastern coast".

Line number: [317-322].

**Comments [37]:** Lines 320-321. Better ". . southern part the flow occupies most of the width of the Red Sea with a narrow northward flow near the eastern coast."

**Reply:** The manuscript is modified as shown in comment 34.

Line number: [317-322].

*"CONCLUSION*

**Comments [38]:** Line 329 to 331 : This sentence should be removed as it is repeated in the last paragraph.

**Reply:** The sentence was removed from manuscript.

**Comments [39]:** Line 332 : « shows »."

**Reply:** The manuscript is modified. Line number: [329].

**Comments [40]:** Line 333. ". . parts show negligible . ."

**Reply:** The manuscript is modified. Line number: [330].

**Comments [41]:** "Line 334 : « geostrophic current ».

**Reply:** The manuscript is modified. Line number: [332].

**Comments [42]:** Line 335-336 : « except along the eastern coast where they flow in the opposite direction ».

**Reply:** The manuscript is modified. Line number: [332-334].

**Comments [43]:** Line 337 : « except a small part […] and of the western coast ».

**Reply:** The sentence was rephrased, as follow.

"In winter, currents flow to the north for the entire sea except for a southward flow along small part of the eastern (22°N-24°N) and western coast (20°N-23°N)".

Line 332 to 336

**Comments [44]:** Lines 341-2 : «. . northern Red Sea more than in the southern part. The mean amplitude . .».

**Reply** The sentence was rewritten, as follow.

". In winter, the cyclonic eddies are beside the western coast and anticyclonic eddies on the eastern side in the Red Sea, while in summer it is concentrated along the central Red Sea in early summer, with some cyclonic eddies transfer to the east coast in late summer".

Line number: [341-344].

**Comments [45]:** Line 341-343 : Please rewrite this sentence."

**Reply:** Please refer to reply to comment (44).

**Comments [46]:** Line 347. Omit "than winter"

**Reply**: The manuscript is modified. Line number: [345].

**Comments [47]:** Line 348. ". . paper is considered . ."

**Reply**: The manuscript is modified. Line number: [346].

Following is the list of response to comments sent by Topic editor, Prof. Huthnance John on 10/1/2019. Some of those comments are already answered in the above responses, the remaining comment are listed below.

**Comments [48]:** Line 163.  "lower" -> "less"

**Reply**: The manuscript is modified. Line number: [162].

**Comments [49]:** Line 181.  ". . FSM-SLA.  White . ."

**Reply**: The manuscript is modified. Line number: [181].

**Comments [50]:** Line 249.  ". . south over almost all the width . ."

**Reply**: The manuscript is modified. Line number: [248].

**Comments [51]:** Line 250.  "anticyclone"

**Reply**: The manuscript is modified. Line number: [249].

**Comments [52]:** Line 251.  "cyclone"

**Reply**: The manuscript is modified. Line number: [249].

**Comments [53]:** Lines 254-5. Not "cause reversed of changes in the direction of flow". Maybe "cause changes in the direction of flow" or "cause reversals of the direction of flow"

**Reply**: The manuscript is modified. Line number: [253-254].

**Comments [54]:** Line 258. ", during this season" -> "in summer".

**Reply**: The manuscript is modified. Line number: [257].

**Comments [55**]: Lines 270-271. I think this is wrong and directly contradicts lines 343-344.

**Reply**: The paragraph is modified in the manuscript as follow.

**"During early summer the eddies are concentrated along the central Red Sea. By August and September some of the cyclonic eddies are shifted towards the eastern coast. During winter the cyclonic eddies are often condensed along the western side of the Red Sea, while anticyclonic eddies along the eastern side of the Red Sea".

Which is now in line with lines 343-344.

Line number: [269-272].

**Comments [56]:** Line 285. More like "3 cm" in figure?

**Reply**: The manuscript is modified. Line number: [286].

**Comments [57]:** Lines 298-299. ". . cyclonic eddy is replaced . ."

**Reply**: The manuscript is modified. Line number: [298].

**Comments [58]:** Line 302. "which" -> "but"

**Reply**: The manuscript is modified. Line number: [301].

**Comments [59]:** Lines 335-336. I think this is wrong and directly contradicts lines 320-321.

**Reply**: This sentence is modified in manuscript. Line number: [316-321] and [331-338]

**Comments [60]:** Lines 341-343. ". . southern side. The mean amplitude for anticyclonic and cyclonic eddies at lower latitudes has small amplitude and at higher latitudes has larger amplitude.

**Reply**: The manuscript is modified. Line number: [339-340].

[revised manuscript text omitted]

---

## Author Response (AR4)

To:

Prof. John M. Huthnance,

Topic Edito

Ocean Science.

Sub:- Submission of revised manuscript [# os-2018-47].

Dear Prof. John M. Huthnance,

Thank you for the reviews and Technical Corrections of our manuscript titled "Estimation of geostrophic current in the Red Sea based on Sea level anomalies derived from extended satellite altimetry data".

We have revised the manuscript in accordance with the technical corrections Topic Editor. The comments really helped us in improving the manuscript further. We thank you and the reviewers for the constructive criticism.

A point-by-point response to the comments of the reviewer is enclosed along with the revised manuscript.

We look forward to hearing from you.

Thanking you,

Yours sincerely,

(A. M. Taqi)

Topic Editor Decision: Publish subject to technical corrections (29 Mar 2019) by John M. Huthnance

Comments to the Author:

Dear Authors

Thank-you for your revisions. I am still asking you to consider many "Technical Corrections" (see below). Three of these marked \* are more serious in that they concern the scientific basis of your findings; you should remember that readers will be able to see all the comments and whether you have properly responded. After this you should enter the manuscript to the Copernicus / Ocean Science production system. It will be copy-edited and you should check that your intended meaning is kept. Thank-you for publishing in Ocean Science.

Yours sincerely

John Huthnance

Technical corrections.

**Technical corrections [1]:** In many places you express a latitude range using "between". This should always be "between m°N and n°N", i.e. always "between . . and . ." This applies in lines 116, 118, 194, 213 (twice), 214, 215, 218, 219, 224, 231, 237, 238, 239, 246, 249, 250, 262, 263, 344 and perhaps others, please check.

Reply: The modified in the manuscript

Technical corrections [2]: Line 22. Delete "the"

Reply: The modified in the manuscript

Technical corrections [3]: Line 47. Delete "S."

Reply: The modified in the manuscript

Technical corrections [4]: Line 48. ". . on a modelling approach"

Reply: The modified in the manuscript

Technical corrections [5]: Line 61. "sea level" (lower case "s")

Reply: The modified in the manuscript

**Technical corrections [6]:** Lines 65-67. Better ". . satellite altimetry which offers large coverage . . SSH (hence sea-level anomaly – SLA), wave height . ."

Reply: The modified in the manuscript

Technical corrections [7]: Line 68. Delete "the"

Reply: The modified in the manuscript

**Technical corrections [8]:** Line 78. ". . (1) echo interference . . as inland water" (delete two "the")

Reply: The modified in the manuscript

Technical corrections [9]: Line 89. ". . important in densely . ."

Reply: The modified in the manuscript

Technical corrections [10]: Line 90. "... regions, ..."

Reply: The modified in the manuscript

Technical corrections [11]: Line 98. ". . Jason-2 altimetry along . ."

Reply: The modified in the manuscript

**Technical corrections [12]:** \*Line 98. "SLA" Anomaly relative to what? For example, is it relative to the mean of the whole 6-year period, or are July anomalies relative to the average of the 6 Julys, or . . ?

Reply: The data used here is the output from Taqi et al, 2017) which based on Jason-2 Satellite altimetry sea level anomaly. This product provides sea-surface-height anomalies

relative to a 16years mean from 1993 through 2009.

Technical corrections [13]: Line 100. "To cover all the period"; what period?

Reply: We meant by that, to extend the data till end of December 2014. The manuscript is modified accordingly.

**Technical corrections [14]:** Lines 102-103. ". . removal from SLA of outliers . . from the mean. . ."

Reply: The modified in the manuscript

Technical corrections [15]: Line 127. "calculating" -> "calculate"

Reply: The modified in the manuscript

Technical corrections [16]: Line 129. "... for comparison ..."

Reply: The modified in the manuscript

**Technical corrections [17]:** Lines 131-132. Better ". . using interpolation (kriging) to smooth the dataset. The merged data are hereafter called FSM-SLA."?

Reply: The modified in the manuscript

Technical corrections [18]: Line 137. "is using" -> "uses"

Reply: The modified in the manuscript

Technical corrections [19]: Line 141. ". . exceed 500"

Reply: The modified in the manuscript

Technical corrections [20]: Line 142. ".. as the level ..."

Reply: The modified in the manuscript

Technical corrections [21]: Line 144. "shown" -> "showed"

Reply: The modified in the manuscript

Technical corrections [22]: Line 145. "is" -> "are"

Reply: The modified in the manuscript

Technical corrections [23]: Line 148. "level" -> "levels"

Reply: The modified in the manuscript

Technical corrections [24]: Lines 149. ". . with root-mean-square error (RMSE) around . ."

Technical corrections [25]: Line 152. Omit "show"

Reply: The modified in the manuscript

Technical corrections [26]: Line 156. Delete first "The"

Reply: The modified in the manuscript

Technical corrections [27]: Lines 161-162. Can now reduce to ". . where the RMSE is less . ."

Reply: The modified in the manuscript

Technical corrections [28]: Line 164. ". . data comparison with observed . ."

Reply: The modified in the manuscript

Technical corrections [29]: Table 2 Note. ". . comparisons is . ."

Reply: The modified in the manuscript

Technical corrections [30]: Line 177. ". . SLA compared with . ."

Reply: The modified in the manuscript

**Technical corrections [31]:** Figure 4 fonts. "October 2011" should match "March 2010" and "April 2011". April longitudes should match March and October longitudes. October colour scale values should match March and April colour scale values. This is for good appearance!

Reply: The modified Figure 4 in the manuscript

Technical corrections [3]2: Line 180. Better ". . FSM-SLA. Green . ."

Reply: The modified in the manuscript

Technical corrections [33]: Line 184. ". . timing of the three cruises . ."

Reply: The modified in the manuscript

**Technical corrections [34]:** Line 187. ". . FSM-SLA with geostrophic currents from CTD data . ."

Reply: The modified in the manuscript

**Technical corrections [35]:** Line 189. ". . currents near the coast estimated from FSM-SLA match"

Reply: The modified in the manuscript

**Technical corrections [36]:** Line 191. ". . AVISO do not always match CTD-derived currents, especially . ."

Reply: The modified in the manuscript

Technical corrections [37]: Line 195. "... while the AVISO ..."

Reply: The modified in the manuscript

**Technical corrections [38]:** Lines 200-201. ". . months March 2010, April 2011 and October 2011 shows . ."

Reply: The modified in the manuscript

**Technical corrections [39]:** Lines 208-209. Better ". . current along the eastern coast of the Red Sea is northward while along the western coast it is southward. . ."

**Reply: The modified in the manuscript**

**Technical corrections [40]:** Line 212. "2002)." Here you start descriptions for each month. Like the others, January should start with a new paragraph "Fig.5 . ." (omit "The").

Reply: The modified in the manuscript

**Technical corrections [41]:** \*Line 212. You need to say something here related to your response about the small standard deviations between the Januarys, between the Februarys etc. as shown in the figure you put in your response. Perhaps an overall standard deviation (average over all months and locations) and locations and magnitude of the largest standard deviations.

Reply: The geostrophic current and eddies from month to month is described from line 212-276

To show that this variability is due to month to month variation the following paragraph is added from 277-285.

"To conform that the above variation is due to month to month variation and not due to the variation between same month from deferent years used for the climatology, a standard deviation between the monthly climatology and months used to create the climatology is estimated. The result show small standard deviation all over the Red Sea for all the months. The highest standard deviation is seen during months of April, October and November with (0.232,0.209 and 0.241) respectively in the northern part along the coast. The lowest standard deviation is seen during March, October and December with (0.008, 0.007,0.009) respectively in the southern part of the Red Sea Show smaller standard deviation than northern part (Figure S1, Table S1). The annual mean standard deviation is about 0.057.

**Technical corrections [42]:** Line 231. ".  $.24^{\circ} - 25^{\circ}$ N and  $20^{\circ} - 22^{\circ}$ N, . ." (delete first ",") Line 232. Delete first ",".

Reply: The modified in the manuscript

Technical corrections [43]: Line 232. ". . seen that several . ."

Reply: The modified in the manuscript

Technical corrections [44]: Line 233. ". . eddies are distributed . ."

Reply: The modified in the manuscript

Technical corrections [45]: Line 235. Delete "the flow of"

Reply: The modified in the manuscript

Technical corrections [46]: Line 235. ". . reversed their direction"

Reply: The modified in the manuscript

Technical corrections [47]: Line 236. "The accompanies formation of a large . ."

Reply: The modified in the manuscript

Technical corrections [48]: Lines 238-239. ". . strength as in May. . ."

Reply: The modified in the manuscript

Technical corrections [49]: Lines 240-241. ". . following the normal . ."

Reply: The modified in the manuscript

Technical corrections [50]: Line 245. Delete "the flow of"

Reply: The modified in the manuscript

Technical corrections [51]: Line 249. ". . Red Sea. Fig. 6 . ."

Reply: The modified in the manuscript

Technical corrections [52]: Line 254. ". . flow. Consequently . ."

Reply: The modified in the manuscript

Technical corrections [52]: Line 256. ". . with a cyclonic eddy, . ."

Reply: The modified in the manuscript

Technical corrections [53]: Line 257. ". . season. Summer . ."

Reply: The modified in the manuscript

Technical corrections [54]: Line 264. ". . towards the south . ."

Reply: The modified in the manuscript

Technical corrections [55]: Line 267. ". . currents is similar . ."

Reply: The modified in the manuscript

Technical corrections [56]: Line 271. ". . Red Sea, with anticyclonic"

Reply: The modified in the manuscript

Technical corrections [57]: Lines 272-273. ". . wind and thermohaline forces (Neumann . ."

Reply: The modified in the manuscript

Technical corrections [58]: Line 275. ". . modified by the presence . ."

Reply: The modified in the manuscript

**Technical corrections [59]:** Line 276. ". . eddies; identification of eddies in the study area was conducted"

Reply: The modified in the manuscript

**Technical corrections [60]:** Lines 277-279. Better "Figure 7 shows latitudinal variability . . . centre of the eddies for 6 years. . ."

Reply: The modified in the manuscript

**Technical corrections [61]:** Lines 280-281. Better ". . stronger than at other latitudes. The amplitude . . defined as the difference between . ."

Reply: The modified in the manuscript

**Technical corrections [62]:** Lines 283-285. "interior. The mean amplitude for an anticyclonic eddy is between 1.3 cm in the southern Red Sea and 5.3 cm in the northern Red Sea and for a cyclonic eddy is between 1.6 cm in the southern Red Sea and 4.2 cm in the northern Red Sea. . ."

**Reply: The modified in the manuscript**

**Technical corrections [63]:** Line 289. ". . about 5-10 cm/s, but reaches three-times greater in the 25°-26°N . ."

Reply: The modified in the manuscript

**Technical corrections [64]:** Line 290. ". . anticyclonic eddies. These results match those observed in a previous study by Zhan . ."

Technical corrections [65]: Line 292. ". . anticyclonic eddies (left"

**Technical corrections [66]:** \*Lines 294, 295, 307. "6-yr mean". You do not have 6 whole years. You must define exactly what mean you are using.

Reply: The modified in the manuscript

Technical corrections [67]: Line 306. "The bias is very small."

Reply: The modified in the manuscript

**Technical corrections [68]:** Line 307. ". . compared with deviation . ." But deviation of what? You must define the variable whose deviation is given in the table.

Reply: The modified in the manuscript

Technical corrections [69]: Line 315. "Figure 9 shows a general schematic . ."

Reply: The modified in the manuscript

Technical corrections [70]: Line 316. ". . north over most of the width of the Red"

Reply: The modified in the manuscript

Technical corrections [71]: Line 317-318. ". . addition, southward geostrophic currents . ."

Reply: The modified in the manuscript

Technical corrections [72]: Line 321. "while in the southern . ."

Reply: The modified in the manuscript

Technical corrections [73]: Line 331. Delete "flow"

Reply: The modified in the manuscript

Technical corrections [74]: Line 334. ". . along a small part . ."

Reply: The modified in the manuscript

Technical corrections [75]: Line 335. ". . study, a northward . ."

Reply: The modified in the manuscript

**Technical corrections [76]:** Line 337. "Cyclonic eddies were relatively larger than anticyclonic eddies . ."

Reply: The modified in the manuscript

**Technical corrections [77]:** Lines 338-339. ". . southern part. Anticyclonic . . small mean amplitude and"

Reply: The modified in the manuscript

Technical corrections [78]: Line 340. ". . larger mean amplitude. . ."

Reply: The modified in the manuscript

Technical corrections [79]: Line 341. "it is" -> "they are"

Reply: The modified in the manuscript

Technical corrections [80]: Line 342. "transferring"

Reply: The modified in the manuscript

Technical corrections [80]: Lines 344-345. ". . 27.5°N. During summer cyclonic . ."

Reply: The modified in the manuscript

Technical corrections [81]: Line 346. ". . polarities were observed . ."

Reply: The modified in the manuscript

Technical corrections [82]: Line 347. "besides"

Reply: The modified in the manuscript

Technical corrections [83]: Line 350. ". . providers: JPL . ."

Reply: The modified in the manuscript

Technical corrections [84]: Line 353. ". . Red Sea. The authors thank the . ."

Reply: The modified in the manuscript

[revised manuscript text omitted]